# ComPC: Completing a 3D Point Cloud with 2D Diffusion Priors

**Tianxin Huang, Zhiwen Yan, Yuyang Zhao, Gim Hee Lee**
School of Computing, National University of Singapore University
`{huangtx, gimhee.lee}@nus.edu.sg`

## Abstract

3D point clouds directly collected from objects through sensors are often incomplete due to self-occlusion. Conventional methods for completing these partial point clouds rely on manually organized training sets and are usually limited to object categories seen during training. In this work, we propose a test-time framework for completing partial point clouds across unseen categories without any requirement for training. Leveraging point rendering via Gaussian Splatting, we develop techniques of Partial Gaussian Initialization, Zero-shot Fractal Completion, and Point Cloud Extraction that utilize priors from pre-trained 2D diffusion models to infer missing regions and extract uniform completed point clouds. Experimental results on both synthetic and real-world scanned point clouds demonstrate that our approach outperforms existing methods in completing a variety of objects. Our project page is at `https://tianxinhuang.github.io/projects/ComPC/`.

## 1 Introduction

3D point clouds have always been an important perceptual approach for the physical world, finding extensive use in various applications such as SLAM (Cadena et al., 2016) or 3D detection (Geiger et al., 2013; Reddy et al., 2018). However, point clouds are often captured from specific camera viewpoints (Yuan et al., 2018; Kasten et al., 2024) in real applications, which may lead to the incompleteness of collected points due to the self-occlusion. Robust completion for partial point clouds can greatly reduce the cost for data collection, and are useful for subsequent 3D perception.

As illustrated in Fig. 1-(a), most existing completion methods (Yuan et al., 2018; Zhao et al., 2021; Zhou et al., 2022; Yu et al., 2023) adopt well-designed deep neural networks to directly generate complete point clouds from partial ones. These methods are usually trained on specific point cloud datasets (Yuan et al., 2018; Yu et al., 2023) and demonstrate outstanding performances on their respective test sets. However, they face challenges in handling data that differs from what they were trained on, such as unseen object categories or real-world scans. This limitation significantly hinders the practical deployment of these point cloud completion methods.

Leveraging the impressive capabilities of 2D diffusion models (Rombach et al., 2022; Saharia et al., 2022; Ho et al., 2020), SDS-complete (Kasten et al., 2024) firstly propose a test-time point cloud completion methods utilizing text-to-3D generative models (Poole et al., 2022; Wang et al., 2023). As shown in Fig. 1-(b), this method optimizes a Neural surface (Yariv et al., 2021) guided by Score Distillation Sampling (SDS) (Poole et al., 2022) of the text-conditioned Stable Diffusion (Rombach et al., 2022). The Neural surface, modeled as a Signed Distance Field (SDF) following VolSDF Yariv et al. (2021), incorporates the geometric details from the partial points by setting their SDF values to zero. The completed points are then generated from the optimized surface for assessment. By tapping into the extensive 2D knowledge provided by diffusion models, SDS-complete (Kasten et al., 2024) manages to achieve significantly robust point cloud completion without any training on specific training sets. However, a notable limitation of the method proposed by SDS-complete (Kasten et al., 2024) is its dependency on manually created text prompts for each point cloud to guide the completion. This requirement can encounter a challenge in real-world applications, where providing detailed and accurate text descriptions for incomplete point clouds is not always feasible.

In view of the above-mentioned issues, we propose a novel test-time point cloud completion framework that eliminates the need for any extra manually provided information such as text descriptions.

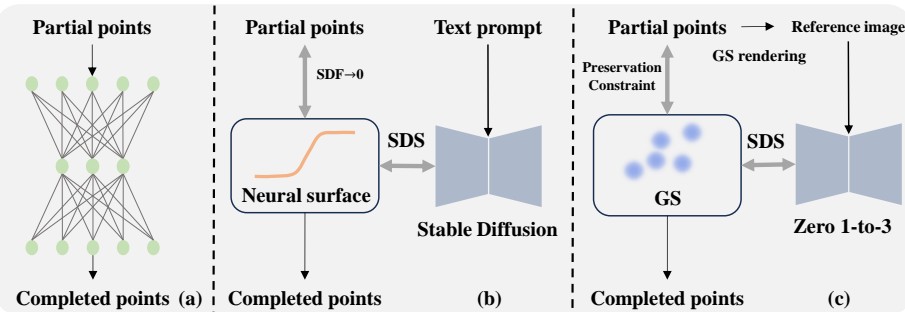

Figure 1: Different point cloud completion methods. (a) Existing network-based completion methods; (b) Test-time SDS-complete (Kasten et al., 2024) with text prompts to guide Neural surface for completion; (c) Our method based on 3D Gaussian Splatting (GS) guided by the diffusion model from Zero 1-to-3 (Liu et al., 2023) conditioned on the reference image rendered from partial points.

As discussed in PCN (Yuan et al., 2018) and SDS-complete (Kasten et al., 2024), existing completion methods concentrate mainly on point clouds incomplete due to self-occlusion, which means that these point clouds often appear nearly complete from at least one viewpoint. Inspired by the amodal perception (Lehar, 1999; Breckon & Fisher, 2005), we aim to complete a point cloud by utilizing the observation from a reference viewpoint that provides the most complete view of the point cloud.

As illustrated in Fig. 1-(c), we estimate such a viewpoint and acquire a reference image of the partial point cloud. Inspired by the capability of novel view synthetic diffusion model, e.g., Zero 1-to-3 (Liu et al., 2023), we propose to use the reference image as a condition for guidance from the diffusion model to infer the missing regions. Utilizing 3D Gaussian Splatting (GS) (Kerbl et al., 2023), which can render 2D images from discrete 3D Gaussians initialized from point clouds, we can effectively render the reference image. This approach also allows us to incorporate 2D diffusion priors into the process of modifying 3D geometry. Consequently, we can complete the missing regions by optimizing the 3D Gaussians with guidance from the 2D diffusion model. Moreover, we propose Preservation Constraint to maintain the geometric integrity of partial point clouds. The completed point clouds would be finally acquired from the 3D Gaussian centers.

Our **main contributions** can be summarized as below:

- We propose the Partial Gaussian Initialization to generate a reference image for partial points, which is observed from an estimated reference viewpoint;

- Based on the reference image, we develop the Zero-shot Fractal Completion to complete the missing regions by introducing 2D diffusion priors;

- We propose Point Cloud Extraction to extract uniform point clouds from 3D Gaussians;

- Through comprehensive evaluation across various data, we demonstrate that our approach surpasses conventional completion methods in handling both synthetic and real-world scanned point clouds.

## 2    RELATED WORKS

### 2.1    3D GENERATION VIA 2D PRIORS

Since the notable success of 2D diffusion models in text-to-image generation (Rombach et al., 2022; Saharia et al., 2022; Ho et al., 2020), text-to-3D and image-to-3D generation have attracted the attention of an increasing number of researchers. To achieve robust and generalizable 3D generation, researchers propose to lift 2D priors for 3D generation (Poole et al., 2022; Wang et al., 2023; Mohammad Khalid et al., 2022; Michel et al., 2022). These works usually optimize specific 3D representations by guidance from 2D diffusion models under different viewpoints, where the guidance is calculated with Score Distillation Sampling (SDS) (Poole et al., 2022) through rendered images. Score Distillation Sampling (SDS) guides a target model (e.g., NeRF) by using gradients from a pre-trained diffusion model. This aligns the target model's output with the diffusion model's learned distribution, enabling high-quality generation in specialized domains.

Zero 1-to-3 (Liu et al., 2023) achieve remarkable 3D generation quality by using SDS guidance from their pre-trained novel view synthesis diffusion model explicitly conditioned on the reference image and camera transformation. Conditioned on a single image, Zero 1-to-3 predicts an image consistent with plausible 3D shapes for any given camera pose. However, its reliance on NeRF representation leads to prolonged optimization times. **3D Gaussian Splatting (GS)** (Kerbl et al., 2023) is an efficient 3D representation that encodes both geometrical and appearance information using a set of 3D Gaussians. Each Gaussian is defined by attributes such as 3D coordinates, scaling, opacity, rotation, and spherical harmonics parameters. By optimizing these attributes, information from 2D images can be incorporated into the Gaussians, enabling efficient novel-view rendering. Dreamgaussian (Tang et al., 2023) offers a solution by optimizing 3D Gaussians through SDS from Zero 1-to-3, achieving a balance between high-quality outputs and acceptable optimization durations.

Motivated by Dreamgaussian, we recognize the potential of GS to refine 3D coordinates of Gaussian centers using guidance from 2D diffusion models. This insight presents an opportunity to apply 2D diffusion priors to tasks related to 3D point clouds, such as point cloud completion.

## 2.2 POINT CLOUD COMPLETION

Point cloud completion aims to recover completed point clouds from partial input point clouds. Ever since PCN (Yuan et al., 2018) firstly applied deep neural networks to predict complete point clouds from partial inputs, numerous advancements (Zhang et al., 2020; Xie et al., 2020; Huang et al., 2020; Yu et al., 2021; Wang et al., 2020; Xiang et al., 2022; Wen et al., 2021) have been made to enhance the accuracy of point cloud completion by altering network architectures. For example, GRNet (Xie et al., 2020) converts point clouds into grid formats and employs 3D CNNs for predicting the completed structures, while PFNet (Huang et al., 2020) adopts a fractal approach to better preserve existing shape details. The Fractal approach focuses on predicting only the missing regions of point clouds, preserving existing details by retaining the shapes from the partial input. RFNet (Huang et al., 2021) utilizes a differentiable layer to merge existing geometrical details from partial point clouds into completed results. More recent approaches (Wang et al., 2024; Zhu et al., 2023; Li et al., 2023; Yu et al., 2021; Xiang et al., 2022; Zhou et al., 2022; Yu et al., 2023; Yan et al., 2022) integrate carefully-designed transformers to improve completion accuracy by considering broader geometric relationships. DiffComplete Chu et al. (2023) is a diffusion-based model for 3D shape completion, leveraging probabilistic modeling to predict missing parts of 3D shapes while preserving structural coherence and diversity.

However, the effectiveness of these point cloud completion methods diminishes when applied to data that differ from their training sets, such as point clouds from unseen categories or other datasets. SDS-complete (Kasten et al., 2024) proposed a test-time completion framework that employs VolSDF (Yariv et al., 2021) for rendering, drawing on priors from pre-trained text-to-image 2D diffusion models (Rombach et al., 2022). This approach maintains the original shapes by constraining the Signed Distance Field (SDF) values of the partial inputs. Yet, this strategy's reliance on text-to-image diffusion models for guidance necessitates well-defined text prompts for each partial point cloud, which may not be practical in real-world applications. Moreover, the optimization of SDS-Complete is quite time-consuming, which may take more than 1000 minutes for one point cloud.

In this study, we propose to leverage 3D Gaussian Splatting (GS) (Kerbl et al., 2023) to bridge point clouds with priors from 2D diffusion models. By generating a reference image of the partial point cloud to serve as a condition for guidance from Zero 1-to-3 (Liu et al., 2023), our method can extract uniform and completed point clouds from the 3D Gaussian centers. Since our method exclusively utilizes information gathered from the incomplete point cloud for completion, it eliminates the need for any additional manually specified prompts for each point cloud. Due to the efficient rendering from 3D GS, and stronger priors from Zero 1-to-3, our method can achieve much higher optimization efficiency than SDS-Complete (Kasten et al., 2024).

## 3 METHODOLOGY

As shown in Fig. 2, the whole completion process is composed of Partial Gaussian Initialization (PGI), Zero-shot Fractal Completion (ZFC), and Point Cloud Extraction (PCE). For the given partial point cloud $P_{in}$, we firstly transform it into colorized reference image $I_{in}$ and 3D Gaussians $G_{in}$

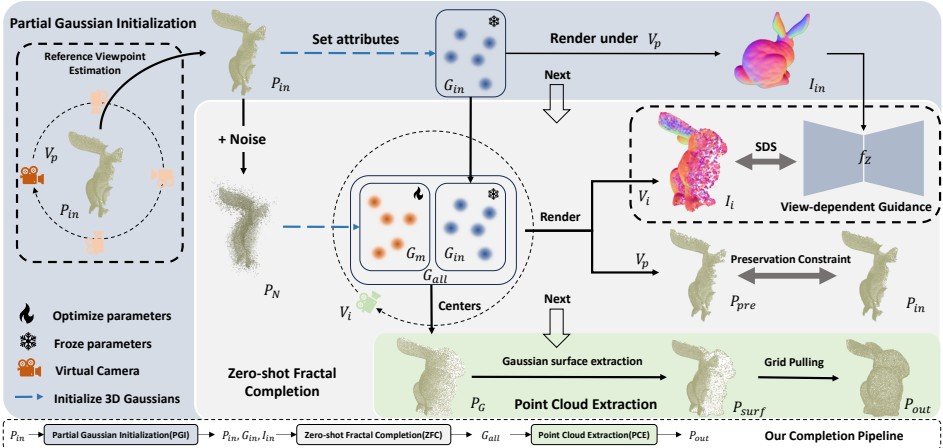

Figure 2: Illustration of our framework. ①In Partial Gaussian Initialization (PGI), Reference Viewpoint Estimation estimates a camera pose $V_p$ where $P_{in}$ can be most completely observed. We initialize 3D Gaussians $G_{in}$ from $P_{in}$ and render the reference image $I_{in}$ under $V_p$. ②In Zero-shot Fractal Completion (ZFC), 3D Gaussians $G_m$ begins with an initialization using noisy $P_N$ and undergoes optimization guided by view-dependent guidance from the diffusion model $f_Z$ in Zero 1-to-3 (Liu et al., 2023) based on a randomly chosen camera pose $V_i$. Additionally, it incorporates a Preservation Constraint computed with respect to $V_p$. $G_{in}$ is mixed with $G_m$ to form $G_{all}$, introducing the partial geometry from $P_{in}$. ③After ZFC, we use Point Cloud Extraction (PCE) to extract surface points $P_{surf}$ from centers of $G_{all}$, and convert $P_{surf}$ into uniform $P_{out}$ with Grid Pulling.

with Partial Gaussian Initialization. Subsequently, $I_{in}$ and $G_{in}$ are introduced to Zero-shot Fractal Completion to acquire 3D Gaussians $G_{all}$ with the completed shape. Specifically, we use $I_{in}$ to guide the optimization of 3D Gaussians $G_m$ by borrowing priors from the 2D diffusion model in Zero 1-to-3 (Liu et al., 2023). Finally, we extract uniform completed point clouds $P_{out}$ from the centers of $G_{all}$ with Point Cloud Extraction. **Please note that the completion is mainly achieved by optimizing 3D Gaussian parameters in** $G_m$, without networks as Yuan et al. (2018).

## 3.1 PARTIAL GAUSSIAN INITIALIZATION

Following the definition of point cloud completion task by PCN (Yuan et al., 2018), only **3D coordinates** are provided as input to infer the complete geometry. To introduce priors from pre-trained 2D diffusion models, we use 3D Gaussian Splatting (GS) to achieve differentiable rendering from 3D point clouds to 2D images. In Partial Gaussian Initialization, we first estimate a reference camera pose $V_p$ with the Reference Viewpoint Estimation. We then initialize 3D Gaussians $G_{in}$ from the incomplete point cloud $P_{in}$. A reference image $I_{in}$ for subsequent completion would be rendered from $G_{in}$ under pose $V_p$. $G_{in}$ is frozen to preserve the geometrical characteristics of $P_{in}$.

**Reference Viewpoint Estimation.** For any point cloud to be completed, we first determine an reference camera pose $V_p$, that captures its most completed observation. The completion process then builds upon this observation. Since the incomplete point cloud $P_{in}$ typically spans across a surface, its most complete view is characterized by minimal self-occlusion and closeness to the camera.

Considering the potential occlusion of rear Gaussians by those in the foreground during rendering, we implement a filter $h(G_{in}, V_n)$ to identify the indices of the frontmost 3D Gaussians in $G_{in}$ from the camera pose $V_n$. Given that the centers of $G_{in}$ are anchored to $P_{in}$, we can estimate $V_p$ by minimizing:

$$V_p = \arg\min_{V_n} \text{CD}(P_{in}[h(G_{in}, V_n)], P_{in}) + w_0 \cdot \text{Depth}(P_{in}, V_n), \tag{1}$$

where $\text{CD}(\cdot, \cdot)$ is the Chamfer Distance (Fan et al., 2017) to measure shape differences between two point clouds. $\text{Depth}(P_{in}, V_n)$ calculates the mean depths of $P_{in}$ observed from the camera at pose $V_n$ for regularization, and $w_0$ is a weighting factor to ensure balance. For this study, we estimate $V_p$ by examining 5,000 camera positions uniformly distributed around the partial point cloud.

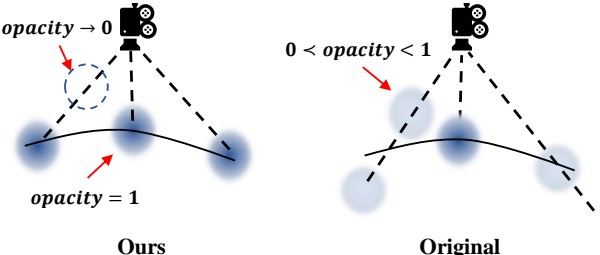

Figure 3: Differences between our binarized opacity and original continuous opacity. $\prec$ denotes smaller but not approaching.

**Gaussian Attributes Setting.** Upon estimating the reference camera pose $V_p$, we render a reference image $I_{in}$ from 3D Gaussians $G_{in}$ initialized from partial point cloud $P_{in}$. To render a characteristic reference image, we make a few modifications to the original 3D Gaussians:

1) The opacity $G_{in}^o$ for all 3D Gaussians within $G_{in}$ is set to a constant value of 1. This step ensures that Gaussians representing all partial points are nearly opaque and clearly visible during rendering.

2) The color $G_{in}^c$ are set as scaled normal map as: $G_{in}^c = (1 + \mathcal{N}(P_{in}))/2$, where the normal vectors $\mathcal{N}(P_{in})$ are estimated with Open3d (Zhou et al., 2018). We scale them from $-1 \sim 1$ to $0 \sim 1$.

## 3.2 ZERO-SHOT FRACTAL COMPLETION

Zero-shot Fractal Completion (ZFC) aims to introduce priors to transform $G_{in}$ with the partial shape into $G_{all}$ with the completed shape. As illustrated in Fig. 2, ZFC optimizes 3D Gaussians $G_m$ for completion and is guided by the View-dependent Guidance and the Preservation Constraint.

**Modification for 3D Gaussians.** 1) Considering point clouds are observed as multiple equal size spheres, we set the scaling of all 3D Gaussians to a single shared scalar value to keep the shape of Gaussians consistent as points. To better cover the space around the partial point cloud $P_{in}$, we create noised $P_N = P_{in} + \mathcal{N}(0, \sigma_n^2)$ for the initialization of $G_m$. The scaling attribute of $G_m$ is initialized as $G_m^s = \frac{1}{|P_N|} \sum Neighbor(P_N)$ from the noisy $P_N$ as shown in Fig. 2, where $Neighbor$ denotes the nearest neighbor distance of each point in $P_N$.

2) Furthermore, as demonstrated in Fig. 3, the original approach to opacity can lead to a dispersion of Gaussian centers around the actual surface due to the range of opacities $0 \prec opacity < 1$ used in rendering. To address this problem, we apply a differentiable quantization (Huang et al., 2022) for Gaussian opacity to binarize the values. For 3D Gaussians $G_m$ with original opacity $G_m^o$, the binarization is implemented as follows:

$$G_m^o = f_{stop}(\text{round}(G_m^o) - G_m^o) + G_m^o, \tag{2}$$

where

$$\text{round}(G_m^o) = \begin{cases} 1 & \text{if } G_m^o > 0.5, \\ \delta & \text{otherwise.} \end{cases}$$

with $f_{stop}(\cdot)$ designed to halt gradient propagation. Here, the forward propagation result of Eq. 2 is $\text{round}(G_m^o)$, while the gradient during backpropagation is calculated based on $G_m^o$. $\delta$ is a predefined small constant set to 0.01 in this work because lower opacity may make the Gaussians hard to optimize. Consequently, 3D Gaussians with $G_m^o \to 1$ cluster near the surface as shown in Fig. 3, while those with $G_m^o \to 0$ will be considered noise and excluded in subsequent processing.

**View-dependent Guidance.** To complete the missing regions, we leverage 2D diffusion priors from Zero 1-to-3 (Liu et al., 2023) due to its capability to deduce the unseen regions based on available imagery. As illustrated in Fig. 2, we utilize the reference image $I_{in}$ from Partial Gaussian Initialization to derive the SDS guidance (Poole et al., 2022) based on image $I_i$ rendered with Gaussian Splatting in a randomly selected viewpoint $V_i$, referred to as View-dependent guidance. Defining $\epsilon_{f_Z}$ as the noise anticipated by the 2D diffusion model $f_Z$ with $t$ and $\epsilon$ indicating the time step and standard noise, respectively, the SDS guidance is calculated as:

$$\nabla_{G_{all}} L_{SDS} = \mathbb{E}_{t,\epsilon}[(\epsilon_{f_Z}(I_i; I_{in}, V_i, t) - \epsilon)\frac{\partial I_i}{\partial G_{all}}]. \tag{3}$$

---

**Algorithm 1** Gaussian Surface Extraction
---

1: Input: 3D Gaussians $G_{all}$ and corresponding centers $P_G$, $h(\cdot)$ following Sec. 3.1
2: **Filtering with opacity:**
3: Let opacity of $G_{all}$ be $G_{all}^o$,
4: Effective 3D Gaussian indexes $id_o = G_{all}^o > 0.5$,
5: **Extracting the surface points:**
6: Set a index list $idx = [\,]$, generate $N$ uniform camera poses $V$,
7: **for** $i = 1$ **to** $N$ **do**
8:   Adding the first observed Gaussian indexes: $idx.\text{append}(h(G_{all}[id_o], V[i]))$
9: **end for**
10: Remove the repeated indexes: $idx = \text{Unique}(idx)$
11: Acquire the surface points: $P_{surf} = P_G[id_o][idx]$

---

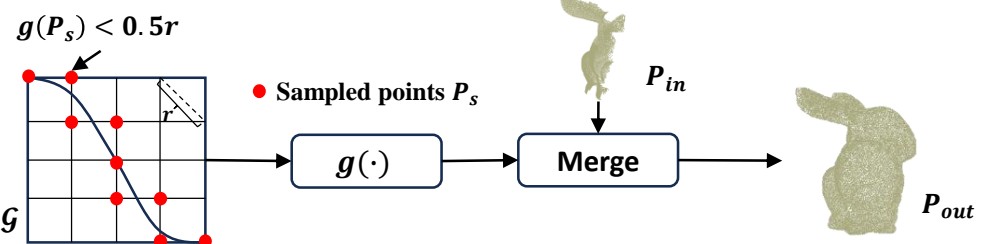

Figure 4: Illustration of Grid Pulling module. $g(\cdot)$ is a MLP-based SDF learned from the completed point cloud $P_{surf}$. Merge denotes merge layer from (Huang et al., 2021). Given the 3D grids $\mathcal{G}$, $r$ is the diagonal length of a unit grid. Sampled points would be $P_s = \{p \mid g(p) < 0.5r, p \in \mathcal{G}\}$.

For the task of point cloud completion, we adopt a fractal approach as discussed in PFNet (Huang et al., 2020) that focuses on optimizing only $G_m$ within $G_{all}$ to reconstruct the missing regions. $G_{in}$ remains unchanged to preserve the original geometric characteristic of the partial point clouds $P_{in}$. Additionally, to manage the scaling $G_m^s$ of 3D Gaussians $G_m$ during optimization, we implement a regularization with a weighting factor of $w_1$:

$$L_{mreg} = w_1 \cdot |G_m^s|. \tag{4}$$

**Preservation Constraint.** To maintain the geometric shapes of the initial partial point clouds, we introduce Preservation Constraint aimed at reducing the shape differences between the partial point cloud $P_{in}$ and Gaussian center coordinates $P_{pre}$ acquired from the partial observation of 3D Gaussians $G_{all}$ under $V_p$. Utilizing the surface filter $h(\cdot, \cdot)$ presented in Sec. 3.1, and considering $G_{all}$ as the combined set of $G_m$ and $G_{in}$ with $P_G[\cdot]$ representing the centers of $G_{all}$, the observed Gaussians centers would be $P_{pre} = P_G[h(G_{all}, V_p)]$. The Preservation Constraint is formulated as:

$$L_p = w_2 \cdot \text{CD}(P_{pre}, P_{in}), \tag{5}$$

where $\text{CD}(\cdot, \cdot)$ is the Chamfer Distance (Fan et al., 2017). $w_2$ is the weighting factor. This constraint ensures the alignment of $G_{all}$ with $P_{in}$ when observed from the reference camera pose $V_p$.

### 3.3 POINT CLOUD EXTRACTION

After the optimization of ZFC, we extract point cloud $P_{out}$ from centers of 3D Gaussians $G_{all}$ with Point Cloud Extraction. Specifically, we firstly select surface points $P_{surf}$ from Gaussian centers $P_G$ with Gaussian surface extraction. We then resample uniform $P_{out}$ from $P_{surf}$ by Grid Pulling.

**Gaussian Surface Extraction.** The centers of the 3D Gaussians can lie both on and below the surface of the shape after optimization. As a result, it is unsatisfactory to directly use these centers as the complete point cloud. To address this issue, we introduce a Gaussian Surface Extraction process to select surface points $P_{surf}$ from the centers of 3D Gaussian $G_{all}$. This procedure is detailed in Alg. 1. By adjusting the opacity of all 3D Gaussians to either $\delta$ or 1, we note that Gaussians with minimal opacity $\delta$ hardly contributes to the rendering process. Consequently, our initial step involves

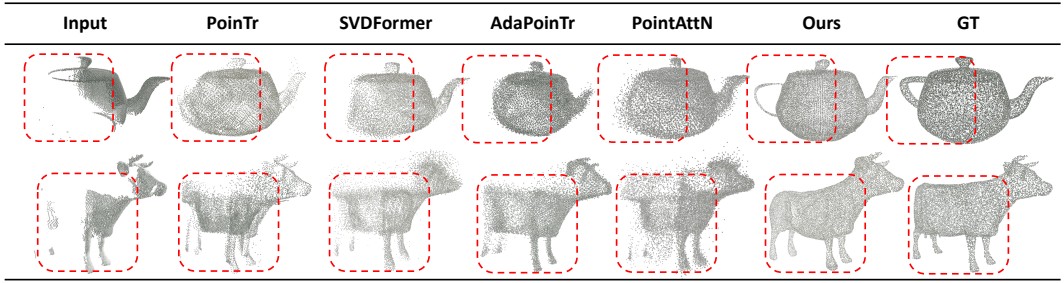

Figure 5: Qualitative comparison on synthetic data.

Table 1: Quantitative comparison on synthetic data. **Bold** marks the best results.

| Object | Horse | MaxPlanck | Armadillo | Cow | Homer | Teapot | Bunny | Nefertiti | Bimba | Ogre | Aver |
|---|---|---|---|---|---|---|---|---|---|---|---|
| Metrics | CD/EMD | CD/EMD | CD/EMD | CD/EMD | CD/EMD | CD/EMD | CD/EMD | CD/EMD | CD/EMD | CD/EMD | CD/EMD |
| PoinTr | 2.75/4.47 | 6.34/6.84 | 3.51/6.07 | 3.13/4.25 | 1.90/4.19 | 3.81/5.12 | 6.39/8.03 | 4.29/5.50 | 5.53/6.73 | 3.41/5.06 | 4.10/5.63 |
| SeedFormer | 3.24/5.30 | 6.91/7.62 | 3.28/6.21 | 3.11/4.00 | 2.04/3.52 | 3.41/4.94 | 6.92/9.10 | 4.25/5.78 | 5.63/7.09 | 3.31/5.73 | 4.21/5.93 |
| PointAttN | 5.25/6.76 | 8.10/8.54 | 5.09/6.65 | 3.73/4.56 | 2.39/3.54 | 5.25/6.36 | 9.35/9.52 | 5.16/5.87 | 8.09/7.52 | 4.80/6.14 | 5.72/6.54 |
| ShapeFormer | 4.17/5.38 | 3.48/4.49 | 3.76/4.68 | 4.53/5.29 | 2.27/2.84 | 2.55/2.86 | 4.52/4.44 | 3.09/3.87 | 5.00/5.85 | 3.39/4.69 | 3.68/4.44 |
| SVDFormer | 2.70/3.89 | 8.37/6.45 | 4.12/6.53 | 3.55/4.39 | 2.42/3.35 | 5.87/6.08 | 6.59/6.90 | 4.27/5.02 | 5.47/4.91 | 4.59/5.36 | 4.79/5.29 |
| AdaPoinTr | 4.88/5.45 | 8.60/8.51 | 5.14/5.95 | 3.48/4.53 | 2.28/3.34 | 3.92/4.56 | 9.33/8.87 | 5.54/6.14 | 8.16/7.64 | 4.53/5.41 | 5.59/6.04 |
| Ours | **0.96/1.32** | **1.23/1.53** | **2.49/4.05** | **1.45/1.64** | **1.34/1.76** | **0.99/1.22** | **1.43/1.78** | **1.81/2.20** | **1.39/1.64** | **1.22/1.67** | **1.43/1.88** |

filtering $G_{all}$ based on opacity as outlined in Alg. 1. To this end, $G_{all}$ is examined from $N$ uniformly distributed camera positions and $h(\cdot, \cdot)$ is employed to extract the centers of the frontmost visible Gaussians as the surface points $P_{surf}$. We set $N = 500$ in this work.

**Grid Pulling.** It is evident in Fig. 2 that the density of points in the completed regions of $P_{surf}$ can significantly differ from that in the original partial point clouds. Ideally, we aim for a consistently dense and uniform distribution of points across the entire shape. Direct attempts to enhance point density within the Zero-shot Fractal Completion (ZFC) would lead to a substantial increase in computational cost. Inspired by NeuralPull (Ma et al., 2020), we introduce a Grid Pulling (GP) module designed to resample points uniformly from initially non-uniform point clouds.

NeuralPull (Ma et al., 2020) employs a Signed Distance Field (SDF) $g(\cdot)$ to pull randomly sampled points $P_{sam}$ that are often generated by adding noise to $P_{gt}$ as $P_{sam} = P_{gt} + N(0, \sigma_0^2)$ towards the surface defined by the original point cloud $P_{gt}$. $\sigma_0$ is the standard deviation for normal distribution $N(0, \sigma_0^2)$. The pulling operation is defined as: $P_{pull} = P_{sam} - g(P_{sam}) \cdot \nabla g(P_{sam}) / \|g(P_{sam})\|_2$. The optimization of $g(\cdot)$ is guided by the Chamfer Distance (CD) as a measure of the distance between $P_{gt}$ and the adjusted points:

$$L_{pull}(P_{sam}, P_{gt}) = \text{CD}(P_{pull}, P_{gt}). \qquad (6)$$

Leveraging $P_{surf}$ obtained from Gaussian Surface Extraction, GP module learns an SDF $g(\cdot)$ to align uniformly sampled points around $P_{surf}$ with its surface. Unlike NeuralPull, which optimizes using only noised point clouds, our approach trains $g(\cdot)$ with both noised point clouds $P_{near} = P_{surf} + N(0, \sigma_0^2)$, and $P_{far}$ being randomly sampled within the 3D bounding box encompassing $P_{surf}$. The loss functions are defined as $L_{far} = L_{pull}(P_{far}, P_{surf})$ and $L_{near} = L_{pull}(P_{near}, P_{surf})$.

Additionally, we utilize a merge layer as suggested by Huang et al. (2021) to incorporate geometric details from $P_{in}$ into $P_{pull}$. Given the distances from $P_{pull}$ points to their nearest neighbors in $P_{in}$ as $dist = \min_{x \in P_{pull}, \forall y \in P_{in}} \|x - y\|_2$, and corresponding neighbor indexes $idx = \arg\min_{x \in P_{pull}, \forall y \in P_{in}} \|x - y\|_2$, the merge layer $g_m$ outputs a set of merged points:

$$g_m(P_{pull}, P_{in}) = e^{-\frac{dist}{\sigma}} P_{in}[idx] + (1 - e^{-\frac{dist}{\sigma}})P_{pull}, \qquad (7)$$

where $\sigma$ is a small optimizable variable to decide how much to merge. The corresponding loss would be $L_{mer} = L_{pull}(g_m(P_{pull}, P_{in}), P_{surf}) + w_3 \cdot \|\sigma\|_2$, where $w_3$ is the weighting factor for the regularization of $\sigma$. The overall training loss for $g(\cdot)$ is then:

$$L_g = L_{far} + L_{near} + L_{mer}. \qquad (8)$$

As depicted in Fig. 4, we initialize a $128^3$ 3D grid $\mathcal{G}$ according to the bounding box of $P_{surf}$. Uniform points $P_s$ would be selected by $P_s = \{p \mid g(p) < 0.5r, p \in \mathcal{G}\}$. $P_s$ is then pulled to

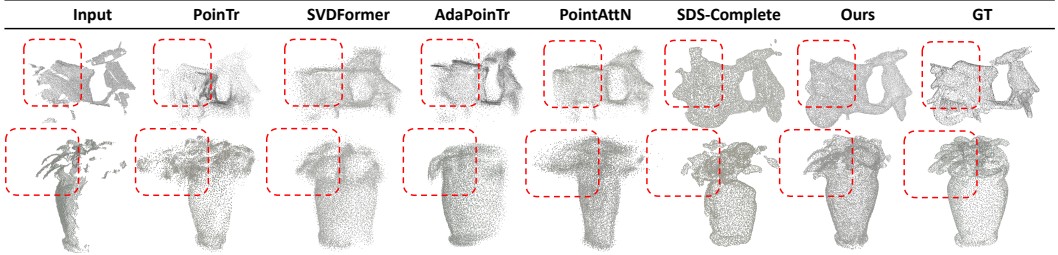

Figure 6: Qualitative comparison on Redwood dataset (Choi et al., 2016; Kasten et al., 2024).

Table 2: Quantitative comparison on Redwood dataset (Choi et al., 2016; Kasten et al., 2024). For the convenience, we re-optimize and normalize the results of SDS-Complete consistently to $-0.5 \sim 0.5$.

| Object | In Domain | | | | | Out Domain | | | | |
| --- | --- | --- | --- | --- | --- | --- | --- | --- | --- | --- |
| | Table | Exe-Chair | Out-Chair | Old-Chair | Average | Vase | Off Can | Vespa | Tricycle | Average |
| Metrics | CD/EMD | CD/EMD | CD/EMD | CD/EMD | CD/EMD | CD/EMD | CD/EMD | CD/EMD | CD/EMD | CD/EMD |
| PoinTr | 3.56/7.42 | 1.91/4.50 | **0.67**/1.41 | 2.48/6.28 | 2.16/4.90 | 3.84/6.55 | 3.76/5.83 | 1.84/3.94 | 5.91/11.63 | 3.84/6.99 |
| SeedFormer | 3.38/7.01 | 1.90/4.55 | 0.76/1.44 | 2.72/5.16 | 2.19/4.54 | 3.99/6.36 | 3.88/6.11 | 2.38/4.38 | 2.10/3.38 | 3.09/5.06 |
| PointAttN | 5.71/7.11 | 2.88/5.65 | 0.73/1.46 | 3.73/6.02 | 3.26/5.06 | 5.35/6.97 | 4.93/6.44 | 2.69/4.70 | **1.72**/3.59 | 3.67/5.43 |
| ShapeFormer | 3.48/5.67 | 3.41/5.32 | 3.87/6.93 | 3.00/4.07 | 3.44/5.50 | 4.79/6.50 | 2.96/3.89 | 3.21/4.20 | 3.21/4.20 | 4.01/5.43 |
| SVDFormer | 2.13/3.29 | 3.60/6.02 | 1.15/2.15 | 3.69/5.83 | 2.64/4.32 | 5.20/7.28 | 5.42/7.05 | 3.30/5.25 | 3.78/4.55 | 4.42/6.03 |
| AdaPoinTr | 5.02/6.23 | 2.58/4.79 | 0.82/**1.38** | 3.62/5.61 | 3.01/4.50 | 5.14/6.48 | 4.47/6.32 | 1.94/3.52 | 1.83/3.67 | 3.34/4.98 |
| SDS-Complete | **1.35/2.30** | 1.96/2.65 | 2.51/3.92 | 2.77/3.77 | 2.15/3.16 | 3.00/5.25 | 3.79/4.28 | 3.36/5.73 | 3.18/3.49 | 3.33/4.69 |
| Ours | 1.67/3.11 | **1.04/1.39** | 1.28/1.73 | **1.42/1.87** | **1.35/2.03** | **2.94/4.63** | **3.51/3.86** | **1.39/2.27** | 2.42/**1.94** | **2.57/3.17** |

the surface of $P_{surf}$ and combined with $P_{in}$ through merge layer. The output point clouds would be $P_{out} = g_m(P_s - g(P_s) \cdot \nabla g(P_s)/\|g(P_s)\|_2, P_{in})$. As $P_{out}$ is quite dense, we sample it to the specified resolution during comparisons.

## 4 EXPERIMENTS

Considering the impracticality of applying test-time completion methods (Kasten et al., 2024) to benchmarks such as Completion3D (Tchapmi et al., 2019) or ShapeNet (Chang et al., 2015) containing thousands of point clouds, we sample an appropriate amount of test data following SDS-Complete (Kasten et al., 2024). For synthetic data, we sample partial point clouds by sampling from various viewpoints around completely modeled objects from established sources (Krishnamurthy & Levoy, 1996; DeCarlo et al., 2003; Praun et al., 2000; Lipman et al., 2008). For real scans, we use Redwood (Choi et al., 2016) following SDS-complete (Kasten et al., 2024). Single scans are used as partial input, while the ground truths are adopted by composing multiple scans. Comparisons on ShapeNet (Chang et al., 2015) and Kitti (Geiger et al., 2013) are presented in the appendix A.

We compare our approach with state-of-the-art supervised methods including PointAttN(Wang et al., 2024), PoinTr (Yu et al., 2021), SVDFormer (Zhu et al., 2023), AdaPoinTr (Yu et al., 2023), SeedFormer (Zhou et al., 2022), ShapeFormer (Yan et al., 2022). As SDS-complete (Kasten et al., 2024) only provides codes for the processing of the Redwood dataset (Choi et al., 2016), we implement corresponding comparisons on Redwood. The evaluation metrics include the L1 Chamfer Distance (CD) and Earth Mover's Distance (EMD) (Fan et al., 2017), which measure the similarity between the reconstructed point clouds and the ground truths. All metrics are multiplied by $10^2$ in subsequent comparisons. We standardize point clouds and perform comparisons at a resolution of 16,384 points following PCN (Yuan et al., 2018). Our results presented for comparisons on both synthetic data and real scans are averaged over three repeated experiments.

### 4.1 COMPARISON ON SYNTHETIC POINT CLOUDS

In this section, we conduct an evaluation on synthetic point clouds. The quantitative and qualitative results are presented in Table 1 and Fig. 5, respectively. Existing network-based methods create noisy and incorrect shapes due to the discrepancies between their training data and the test data. As shown in Fig. 5, our method creates correct and reasonable completed results, which may benefit from abundant

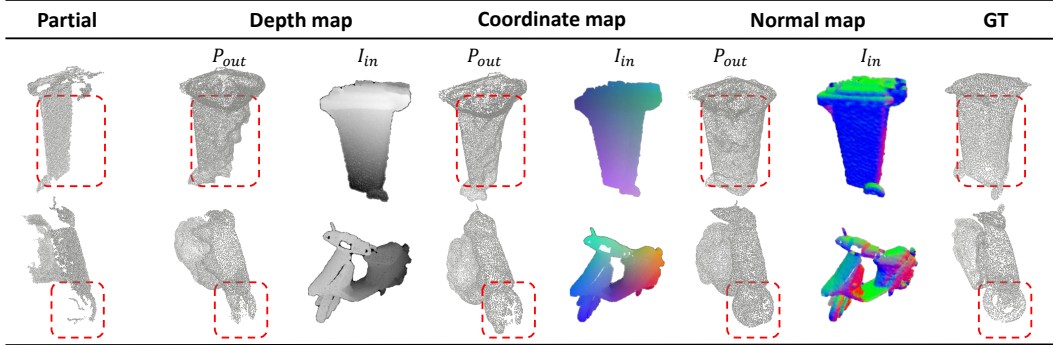

| Partial | Depth map | | Coordinate map | | Normal map | | GT |
|---|---|---|---|---|---|---|---|
| | $P_{out}$ | $I_{in}$ | $P_{out}$ | $I_{in}$ | $P_{out}$ | $I_{in}$ | |

Figure 7: Qualitative comparison between different colorization strategies. $I_{in}$ and $P_{out}$ denote the colorized reference image and completed point clouds, respectively.

Table 3: Ablation for colorization.

| | Depth | Coordinates | Normal(Ours) |
|---|---|---|---|
| CD | 2.25 | 2.01 | **1.96** |
| EMD | 2.88 | 2.64 | **2.60** |

Table 4: Ablation for ZFC and PCE.

| Guidance | Pres | Surf | GP | Redwood CD | Redwood EMD | Synthetic CD | Synthetic EMD |
|---|---|---|---|---|---|---|---|
| ✓ | | | | 1.98 | 3.20 | 3.35 | 6.01 |
| ✓ | ✓ | | | 1.97 | 3.07 | 2.55 | 4.41 |
| ✓ | ✓ | ✓ | | **1.50** | 3.38 | **1.17** | 3.74 |
| ✓ | ✓ | ✓ | ✓ | 1.96 | **2.60** | 1.43 | **1.88** |

priors from the pre-trained diffusion model. An interesting case is that our method completes an appropriate handle for the teapot in the first row of Fig. 5 without any prompts and related geometries. It confirms that the pipeline can actually percept the actual categories of completed objects instead of simply inferring a shape to fill in the missing regions.

## 4.2 COMPARISON ON REAL SCANS

We follow SDS-complete (Kasten et al., 2024) for the comparison on real scans from Redwood (Choi et al., 2016). Scans are divided into the "in domain" categories similar as training datasets of existing completion networks (Yu et al., 2021; Zhou et al., 2022; Yu et al., 2023; Wang et al., 2024), and "out domain" categories unseen during their training. The qualitative and quantitative comparison results are illustrated in Fig. 6 and Table 2, respectively. As shown in Table 2, our method outperforms other methods on both "in domain" and "out domain" models, which further confirms the effectiveness and generalizability of our method. Existing fully-supervised methods may perform inferior even on the in-domain objects as illustrated in Table 2, which reveals their limitation on datasets differing from the training one. By introducing abundant priors from 2D diffusion model (Liu et al., 2023), our method can achieve robust completion for objects across different datasets.

## 4.3 ABLATION STUDY FOR COLORIZATION STRATEGIES IN PGI

To confirm the necessity of using normal map for colorization in Partial Gaussian Initialization, we compare their performances against other strategies including using depth values and normalized coordinates. As shown in Fig. 7, these alternative strategies are clearly outperformed by the normal map composed of normal vectors, particularly in the circled areas. This superiority likely stems from the ability of normal vectors to more distinctly reflect surface changes in colors, thus better capturing the geometric characteristics in the reference image. We also provide quantitative comparisons of different colorization strategies in Table 3, using average metrics from in-domain and out-of-domain Redwood dataset. The results show that the normal map consistently outperforms other methods.

## 4.4 ABLATION STUDY FOR ZFC AND PCE

In this work, we propose ZFC to introduce diffusion priors to infer the missing regions, and PCE to extract uniform point clouds from the 3D Gaussian centers. ZFC is composed of view dependent guidance and Preservation Constraint, while PCE consists of Gaussian surface extraction and Grid Pulling. From Fig. 8, we can see that our method with all components have uniform and reasonable

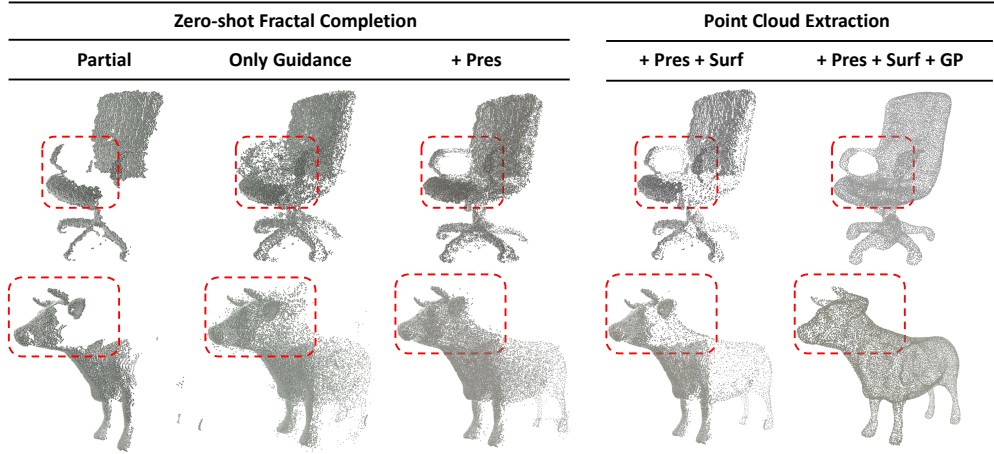

Figure 8: Qualitative ablation study for ZFC and PCE. Surf, Pres, and GP denote Gaussian Surface Extraction, Preservation Constraints, and Grid Pulling, respectively.

completed results. In PCE, GP obviously generates quite uniform point clouds from the non-uniform ones directly acquired from 3D Gaussians. Gaussian Surface Extraction operation extracts the surface from relatively disorganized Gaussian centers. In ZFC, view dependent guidance creates coarse results with relatively correct overall shapes. Preservation Constraint avoids redundant shapes by introducing strict constraints between partially observed points and existing partial point clouds.

We also provide quantitative ablation study for our proposed components in Table 4. We evaluated our method on both Redwood and synthetic datasets. The results demonstrate that the Preservation Constraint improves performance compared to standard view-dependent diffusion guidance. Although Gaussian surface extraction significantly enhances the CD metric by selecting surface points, it negatively affects the EMD metric due to the high non-uniformity, as shown in the fourth column of Fig. 8. In contrast, the final Grid Pulling (GP) module acquire more uniform surface points, leading to better EMD performance, although the CD metric experiences a slight decline due to precision loss caused by potential deformations in GP. More detailed ablation study can be found in the appendix A.

## 5 CONCLUSION

In this work, we introduce a test-time point cloud completion framework that leverages the rich priors from 2D diffusion models (Liu et al., 2023; Zhang et al., 2023) through 3D Gaussian splatting, which can robustly complete partial 3D point clouds without any training requirements. Our framework consists of three main components: Partial Gaussian Initialization (PGI) to render the reference image, Zero-shot Fractal Completion (ZFC) to complete the shape, and Point Cloud Extraction (PCE) to extract point clouds. Our method outperforms both existing network-based and test-time approaches in achieving robust completion across multiple categories of both synthetic and real scanned data.

## LIMITATION

Our method shares similar limitations as claimed by SDS-complete (Kasten et al., 2024). As a test-time completion method, although our method does not require any training, the optimization on the test data would take relatively long time cost. For instance, completing a point cloud from the Redwood dataset takes approximately 15 minutes with our method on a RTX A6000 GPU. However, our framework is much more efficient than the existing test-time method SDS-complete (Kasten et al., 2024), which takes up to 1950 minutes for optimization as reported in their supplementary material.

## ACKNOWLEDGEMENT

This research / project is supported by the National Research Foundation (NRF) Singapore, under its NRF-Investigatorship Programme (Award ID. NRF-NRFI09-0008), and the Agency for Science, Technology and Research (A*STAR) under its MTC Programmatic Funds (Grant No. M23L7b0021).

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

## A APPENDIX

### A.1 DETAILS OF PARAMETER SETTINGS

In Table 5, we provide detailed information on the hyper-parameters discussed in Sec. 3. Our experiments are conducted on RTX A6000/A5000 GPU, with PyTorch 1.12 and CUDA 11.6.

Table 5: The setting of mentioned hyper-parameters in Sec. 3.

|  | Hyper-parameters |
| --- | --- |
| $w_0 \sim w_3$ | 1e-3, 1e3, 1e2, 0.1 |
| $\delta, \sigma_0, \sigma_n$ | 0.01, 0.005, 0.05 |
| Iterations | 1000 (ZFC), 5000 (PCE) |

**Ablation for Grid Pulling.** Grid Pulling (GP) module is proposed to resample uniform and regular point clouds from non-uniform $P_{surf}$ in the Point Cloud Extraction. As claimed in Sec. 3.3, $L_{far}$ and $L_{near}$ are used to optimize an continuous surface presented by MLP while Merge layer is introduced to merge output point clouds with partial input. From results in Fig. 9, we can observe that $L_{far}$ and $L_{near}$ contribute to overall contours and local shapes, respectively. Nonetheless, they are still limited to the over-smoothed results. The merge layer helps preserve local geometrical details in the circled regions from the partial input point cloud. We also provide a quantitative comparison on GP module in Table 6. We can see that each component in GP contributes to the final performance.

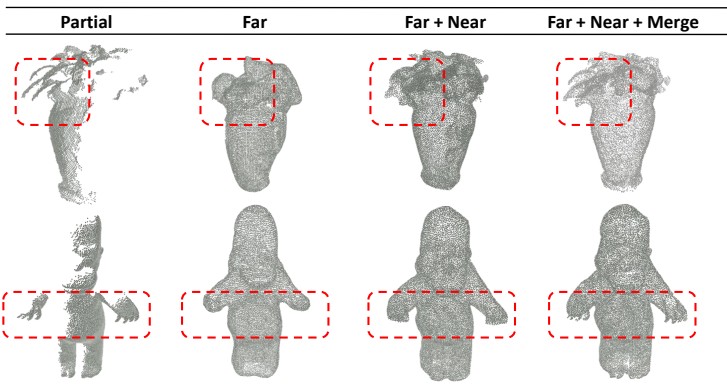

Figure 9: Ablation study for Grid Pulling module. Far, Near, and Merge denote the $L_{far}$, $L_{near}$, and merge layer $g_m(\cdot)$, respectively.

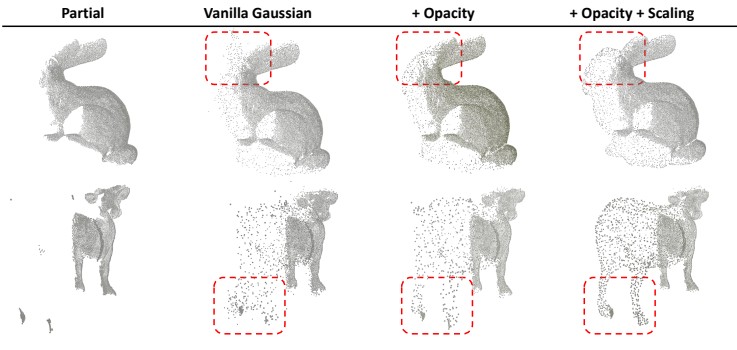

Figure 10: Ablation Study for the 3D Gaussian modifications. Scaling and Opacity denotes the parameter-shared scalar scaling and binary opacity operations mentioned in Sec. 3.2, respectively.

### A.2 ABLATION STUDY FOR MODIFICATIONS OF 3D GAUSSIANS

As presented in Sec. 3.2, we make a few modifications to the original 3D Gaussian Splatting (Kerbl et al., 2023) including the parameter-shared scalar scaling definition and binary opacity estimation. In

Table 6: Quantitative comparison for Grid Pulling module evaluated on Redwood dataset.

|  | No Merge & Near | No Merge | Ours |
|---|---|---|---|
| CD | 3.11 | 2.04 | **1.96** |
| EMD | 3.11 | 2.64 | **2.60** |

Table 7: Quantitative comparison for 3D Gaussian modifications evaluated on Redwood dataset.

|  | No Opacity & Scaling | No Scaling | Ours |
|---|---|---|---|
| CD | 6.35 | 2.50 | **1.96** |
| EMD | 10.48 | 3.58 | **2.60** |

this section, we conduct a few experiments to validate the effectiveness of these proposed operations. To better illustrate their performances, we conduct comparisons based on $P_{surf}$ directly acquired from the Gaussian centers in ZFC. The results are presented in Fig. 10.

Adopting a shared scalar scaling helps in revealing more defined geometric details in the point cloud completion task. The original settings of separate scaling across different 3D Gaussians tend to produce blurring edges and lose finer details. In addition, the binary opacity operation obviously reduce the noises in $P_{surf}$. With the original opacity settings, a considerable number of 3D Gaussians with moderate opacity values would scatter around the actual surfaces, blurring the distinction between the object and its surroundings. The binary opacity method effectively eliminates this issue, ensuring a cleaner bounding and more accurate surface representation. As shown in Table 7, the modifications on 3D Gaussians have significant influence on the completion performances.

### A.3 EFFECT OF THE FRACTAL COMPLETION STRATEGY

As illustrated in Fig. 2, we introduce the fractal completion strategy in ZFC by optimizing 3D Gaussians $G_m$ together with frozen 3D Gaussians $G_{in}$ initialized from partial point clouds. In this section, we conduct experiments to verify the effect of this strategy. A few visualized examples are presented in Fig. 11. When not using fractal completion strategy, we directly optimize 3D Gaussians $G_m$ for all structures without concatenation with $G_{in}$. We observe that completions without the fractal completion strategy tend to overlook some shape details present in the input partial point clouds. The quantitative results in Table 8 further validate the advantages of the fractal strategy.

### A.4 FAILURE CASES

Fig. 12 presents some failure cases. Our method encounters similar problems as SDS-complete (Kasten et al., 2024) when generating thin surfaces in occluded areas. In these cases, 2D diffusion priors tend to imagine the thin occluded regions as reasonable but thicker structures as shown in Fig. 12. This problem could be potentially addressed by fine-tuning the 2D diffusion priors, or introducing some regularization during the optimization process. We will explore it in our future work.

### A.5 DISCUSSION ABOUT NOISY REFERENCE OBSERVATION

Partial point clouds may also be affected by noise due to poor natural illumination or reflections from object surfaces. To evaluate the robustness of our method to such noise, we introduce varying levels of noise to the synthetic partial point clouds described in Sec.4. The quantitative results are summarized in Table 9, and qualitative comparisons are shown in Fig. 13. While the performance of our method decreases as the noise level increases, it consistently outperforms existing approaches. As illustrated in Fig. 13, noise with a standard deviation of 0.01 introduces noticeable blurring to the input partial points, yet our method is still able to recover the overall contour effectively. This demonstrates that our approach exhibits a degree of robustness to noise. The primary contribution of this work lies in the development of a practical framework that leverages 2D diffusion priors for 3D point cloud completion. Comparisons on real scans from the Redwood dataset Choi et al. (2016) validate the effectiveness of our method in handling real-world data. Enhancing its robustness may further remain a promising direction for future research.

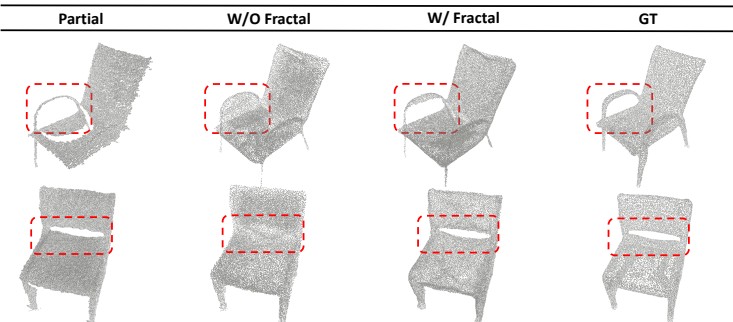

Figure 11: Ablation Study for the Fractal completion strategy. W/ Fractal and W/O Fractal denote using and not using Fractal completion strategy, respectively.

Table 8: Quantitative comparison for the Fractal strategy evaluated on Redwood dataset.

|  | w/o Frac | w/ Frac |
|---|---|---|
| CD | 2.00 | **1.96** |
| EMD | 2.69 | **2.60** |

## A.6 Discussion about different Incompleteness levels

In this section, we evaluate the performances of our method on partial input with different incompleteness levels. For convenience, we use synthetic objects from Sec. 4 to construct evaluation sets with varying levels of incompleteness. Specifically, we initialize the first virtual camera at a pose of $elevation = 0$, $azimuth = -140$, and $fov \approx 80°$. Additional virtual cameras are placed along the azimuth at $15°$ intervals. By merging 1, 3, and 7 consecutive depth maps, we generate partial point clouds with different levels of incompleteness. These data are used for comparison experiments. The qualitative and quantitative comparisons are presented in Fig. 14 and Table 10, respectively. We can see that more completed partial input constructed from more depth maps will bring finer details to the completed results. Our method consistently outperforms other methods in this setting.

## A.7 Evaluation on Multi-modal metrics

Since our method relies on SDS guidance from Zero 1-to-3 Liu et al. (2023), it may produce different completion results with each optimization. To evaluate its performance under these variations, we assess our method using multi-modal metrics, including TMD, UHD, and MMD, following the approach in (Chou et al., 2023). We perform four repeated optimizations for both our method and SDS-complete on the in-domain categories of Redwood, as detailed in Sec. 4.2, to compute these metrics. The results are summarized in Table 11. Our method achieves superior performance on UHD and MMD metrics, further validating its effectiveness for 3D point cloud completion. Although it shows a lower TMD, which evaluates completion diversity, this actually reflects its steady convergence toward the ground truths—a positive attribute for the task of 3D point cloud completion.

## A.8 Comparisons based on Meshes

Our method potentially support the generation of 3D meshes due to the introducing of Grid Pulling module. As mentioned in Sec 3, the Grip Pulling is proposed to re-sample uniform points from the non-uniformed point cloud $P_{surf}$, where a SDF fuction $g(\cdot)$ is introduced to fit the overall shape of $P_{surf}$ to do the resampling. Therefore, we can use Marching Cubes following NeuralPull Ma et al. (2020) to extract meshes from $g(\cdot)$. The results are presented in Fig. 15. We can see that our method can also create more accurate mesh shapes than SDS-Complete Kasten et al. (2024).

## A.9 Evaluation on ShapeNet

In this section, we further compare our methods with network-based methods on 16 common models from 4 different categories of ShapeNet dataset. The results are presented in Table 12 and Fig. 16. Although our method performs slightly inferior to network-based methods on the known category

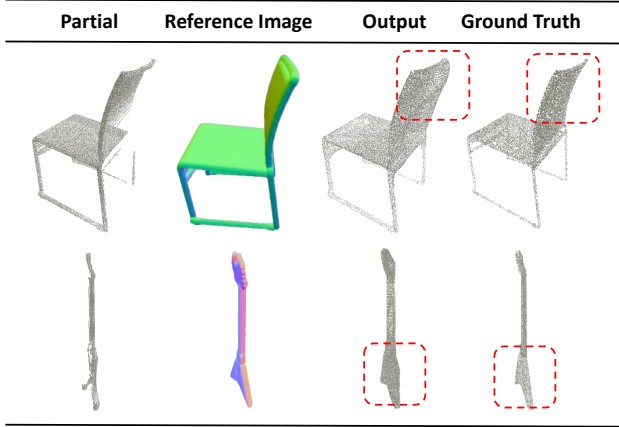

Figure 12: Some failure cases.

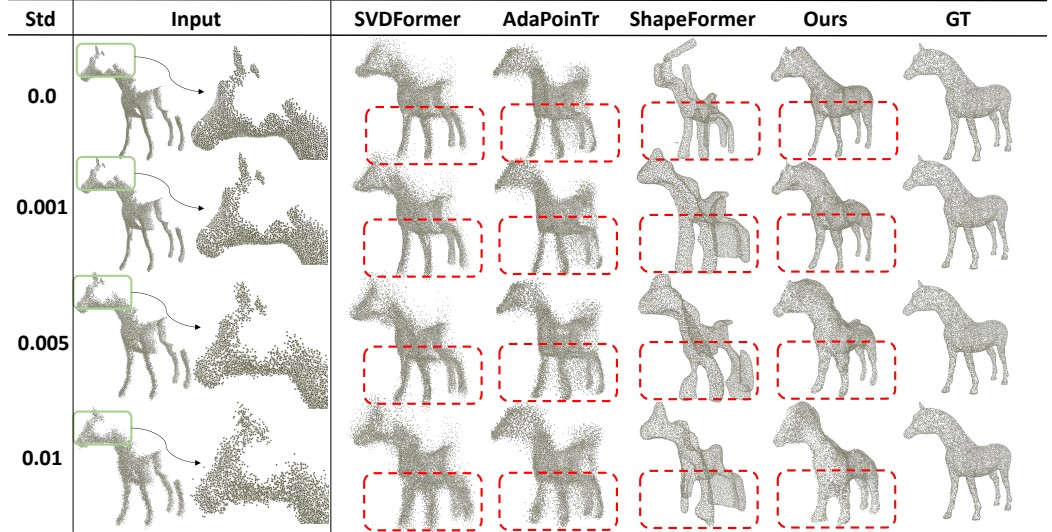

Figure 13: Qualitative comparisons under different noise perturbations. Std denotes the Standard deviation of added noises. The green box marks a local area of a noised point cloud.

objects, it surpasses other methods on the unknown category objects. Please note that network-based methods use 3D ground truths from known categories for supervision during training, while our method does not introduce any training with such ground truths. As a test-time point cloud completion method, the core contribution of our method is its generalizablity for point cloud objects from any category. This has been confirmed by experiments on multiple types of data including synthetic objects in Sec. 4.1, Redwood dataset in Sec. 4.2, and ShapeNet in Sec. A.9.

## A.10 EVALUATION ON LIDAR POINTS

As discussed in Sec. 3, we render the reference image $I_{in}$ from the incomplete point cloud $P_{in}$, under the estimated camera pose $V_p$. This operation means that we actually observe the point cloud from a pinhole camera model, which may be closer to point clouds from depth scanners, such as Redwood dataset (Choi et al., 2016; Kasten et al., 2024). To validate the effectiveness of our method across different sensor types, we conduct a comparison using point clouds from the Kitti dataset (Geiger et al., 2013), which are acquired with LiDAR sensors. Point clouds from Pedestrian, Cyclist, Car, and Truck are adopted for evaluation. Since ground truth data are unavailable for these point clouds, we mainly present qualitative comparison in Fig. 17. Notably, our method demonstrates the ability for reasonable completion even with LiDAR-derived point clouds.

Table 9: Quantitative comparisons on noised input point clouds. Std denotes the Standard deviation of added noises.

| Std | PoinTr CD/EMD | Seedformer CD/EMD | PointAttN CD/EMD | SVDFormer CD/EMD | ShapeFormer CD/EMD | AdaPoinTr CD/EMD | Ours CD/EMD |
|---|---|---|---|---|---|---|---|
| 0 | 4.10/5.63 | 4.21/5.93 | 5.72/6.54 | 4.79/5.29 | 3.68/4.44 | 5.59/6.04 | **1.43/1.88** |
| 0.001 | 4.15/5.59 | 4.17/5.91 | 5.76/6.55 | 4.73/5.22 | 3.65/4.52 | 5.59/6.04 | **1.53/1.87** |
| 0.005 | 4.24/5.83 | 4.31/6.11 | 5.75/6.59 | 4.92/5.52 | 4.03/4.89 | 5.65/6.19 | **2.02/2.25** |
| 0.01 | 4.16/5.85 | 4.34/6.22 | 5.62/6.73 | 4.86/5.75 | 4.06/5.00 | 5.57/6.44 | **3.18/4.07** |

Table 10: Quantitative comparisons under different incompleteness levels. The levels denote how many depth maps are used to construct the partial input.

| Level | PoinTr CD/EMD | Seedformer CD/EMD | PointAttN CD/EMD | SVDFormer CD/EMD | ShapeFormer CD/EMD | AdaPoinTr CD/EMD | Ours CD/EMD |
|---|---|---|---|---|---|---|---|
| 1 | 3.77/5.13 | 4.16/6.02 | 5.52/6.29 | 4.63/5.08 | 3.30/4.07 | 5.33/5.82 | **1.86/2.01** |
| 3 | 3.61/5.10 | 3.92/5.93 | 5.45/6.28 | 4.36/5.02 | 3.42/4.26 | 5.28/5.82 | **1.76/2.04** |
| 7 | 3.06/4.95 | 3.48/5.72 | 5.18/6.13 | 4.16/4.95 | 3.00/3.77 | 5.26/5.85 | **1.48/1.87** |

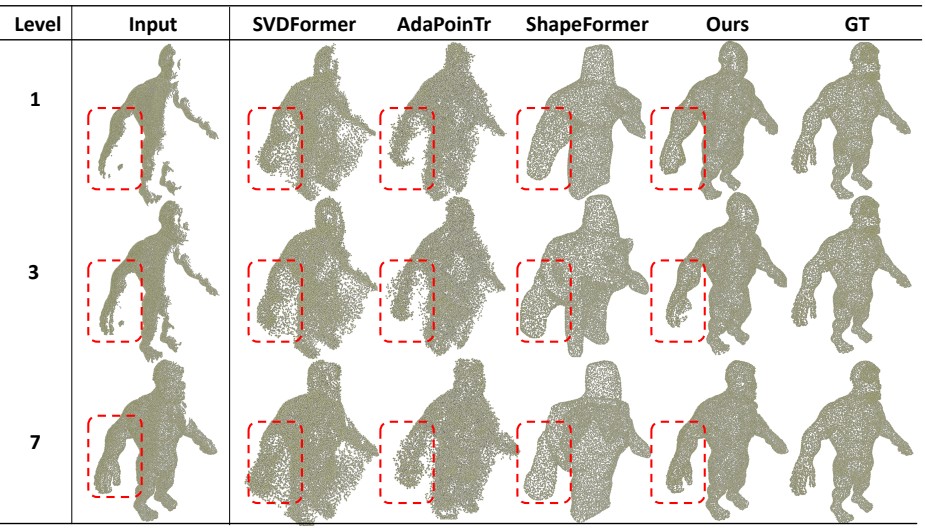

Figure 14: Qualitative comparisons under different incompleteness levels. The levels denote how many depth maps are used to construct the partial input.

Table 11: Quantitative comparisons on multi-modal metrics.

| Methods | Metrics | Table | Exe-Chair | Out-Chair | Old-Chair | Aver |
|---|---|---|---|---|---|---|
| SDS-Complete | TMD ↑ | **1.26** | **1.70** | **1.18** | **1.25** | **1.35** |
| | UHD ↓ | 9.31 | 10.39 | 10.63 | 16.67 | 11.75 |
| | MMD ↓ | **1.27** | 1.66 | 1.86 | 2.11 | 1.73 |
| Ours | TMD ↑ | 0.57 | 0.53 | 0.42 | 0.65 | 0.54 |
| | UHD ↓ | **8.47** | **4.73** | **8.64** | **12.02** | **8.47** |
| | MMD ↓ | 1.47 | **1.04** | **1.28** | **1.42** | **1.30** |

Table 12: Quantitative comparison on ShapeNet dataset. "Known category" and "Unknown category" denote categories included and not included in the training set of network-based methods, respectively.

| Categories | Known category | | Unknown category | |
|---|---|---|---|---|
| | Chair | Table | Pistol | Tower |
| Metrics | CD/EMD | CD/EMD | CD/EMD | CD/EMD |
| PoinTr | **1.31**/2.64 | **0.74**/2.86 | 1.84/3.84 | 2.38/3.05 |
| SeedFormer | 1.39/2.77 | 0.80/2.17 | 1.79/3.91 | 1.95/3.24 |
| AdaPoinTr | 1.45/2.54 | 0.74/1.58 | 2.28/4.15 | 1.99/3.30 |
| PointAttN | 1.26/2.56 | 0.92/1.93 | 2.48/4.83 | 1.72/3.03 |
| SVDFormer | 1.21/2.49 | 1.68/3.15 | 2.02/4.25 | 3.47/4.22 |
| Ours | 1.38/**1.94** | 1.08/**1.56** | **1.09/1.63** | **1.41/1.82** |

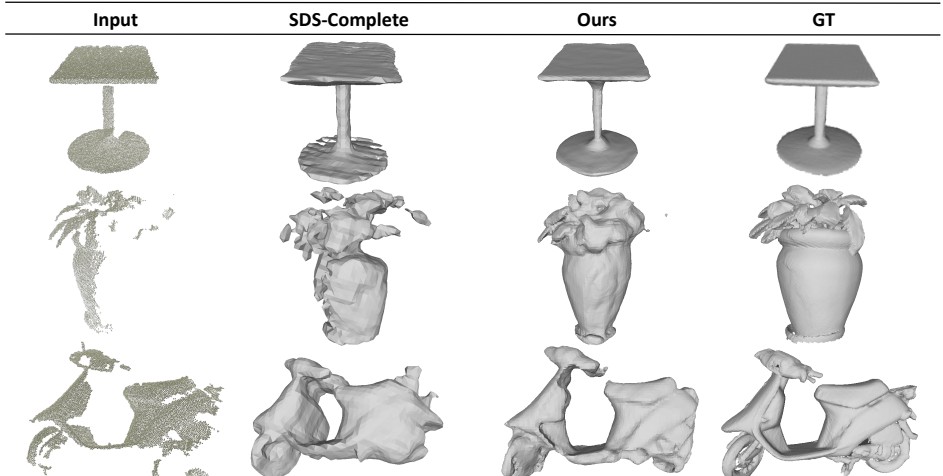

Figure 15: Comparisons based on Meshes.

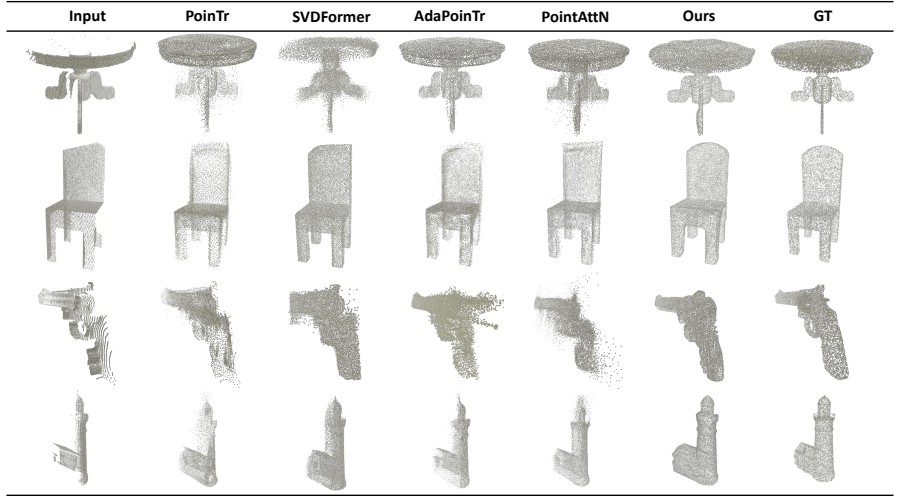

Figure 16: Qualitative comparison on objects from ShapeNet (Chang et al., 2015) dataset.

| Input | PoinTr | AdaPoinTr | SVDFormer | PointAttN | Ours |
|---|---|---|---|---|---|

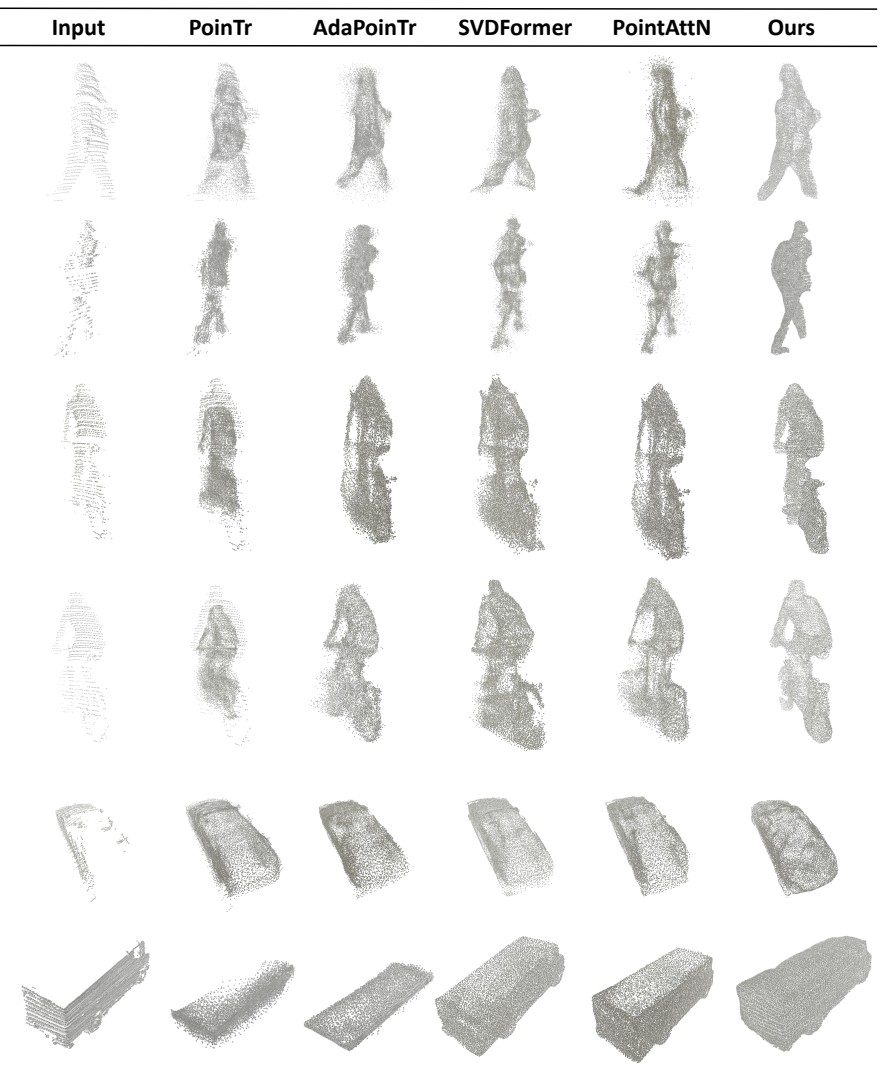

Figure 17: Comparison on Kitti (Geiger et al., 2013) dataset.

