# OpenReview forum: "ComPC: Completing a 3D Point Cloud with 2D Diffusion Priors"
_ICLR.cc/2025/Conference — ICLR 2025 Poster_

### Official Review · Reviewer_p7jk · 2024-10-19

**Soundness:** 3
**Presentation:** 3
**Contribution:** 3
**Rating:** 8
**Confidence:** 3

**Summary:**

The work tries to complete a general partial point cloud using  zero123, which gets an object's image and outputs images of the exact object from any unseen direction.
They first model the partial point cloud as gaussians, and render them from the view direction, where the object seems full.
than through SDS iterations using zero123 they complete the object from any noval direction.

**Strengths:**

They continue works that complete point clouds in the wild and challenge the current weakness of long-running time and the need for textual description.
The work exploits advances in 3D generation for further improvement in point cloud completion.

**Weaknesses:**

good work

**Questions:**

1. I can't see how your work can generate a surface like SDS-complete?

---

> ### Author Response · Authors · 2024-11-22
> **Response to Reviewer p7jk**
>
> **Q1**: I can't see how your work can generate a surface like SDS-complete?
>
> **A1**: Thank you for liking our work! In this work, we focus on the task of point cloud completion, as commonly defined in prior studies such as [1, 2], which involves recovering complete 3D point clouds from partial inputs. Generating 3D meshes is not the primary goal of this task, so we base our comparisons on point clouds.
>
> However, our method has the potential to generate 3D meshes thanks to the introduction of the Grid Pulling module. As described in Sec. 3.3, Grid Pulling re-samples uniform points from the non-uniform point cloud $P_{surf}$ by fitting a signed distance function (SDF) $g(\cdot)$ to the overall shape of $P_{surf}$. Using Marching Cubes, as outlined in [3], meshes can be extracted from $g(\cdot)$. Qualitative results, presented in  **Sec. A.7 of the appendix**, show that our method can produce higher-quality completed meshes compared to SDS-complete.
>
> [1] PCN: Point Completion Network
> [2] GRNet: Gridding residual network for dense point cloud completion.
> [3] Neural-pull: Learning signed distance functions from point clouds by learning to pull space onto surfaces.

---

### Official Review · Reviewer_y9Sz · 2024-11-01

**Soundness:** 3
**Presentation:** 3
**Contribution:** 4
**Rating:** 8
**Confidence:** 4

**Summary:**

This paper proposes a new pipeline for point cloud completion. The proposed pipeline consists of three steps. The pipeline initializes a single optimal viewpoint for initial render of point cloud. This render is made so the point cloud is most completely observed. The Gaussian Splatting is used to initialize a frozen set of 3D gaussians by utilizing an optimal viewpoint. In the next step the set 3D gaussians are prepared in the optimal way to cover missing point cloud regions. Further Zero 1-to-3 approach is used to complete missing parts of the point cloud. On the final step the point cloud is extracted from the 3D gaussians distributed uniformly (unlike in the start of pipeline) on the shape surface.

Despite the weaknesses, I recommend accepting this work because the authors definitely introduced a new strong pipeline that incorporates many existing methods. Authors proved that the proposed sequence of methods is successfully used for the important complicated task.

The main reason to accept this work is good results that superior existing point cloud completion methods based on single-modality input data. The method shows good performance in the training domain and out of the training domain. In some cases (depicted in the paper) the density of points seems to be more uniform than in GT point clouds. This feature can be useful in tracking and segmentation tasks.

**Strengths:**

Learning-based methods perform not good when tested out of the domain they were trained on. The strong point of the proposed method is that authors propose a framework that performs out of the training domain as good as in it. Moreover, the model does not require any additional modalities, so the possibility of application is maximal. Lastly, there are empirical experiments done that prove that proposed method outperforms overviewed analogical approaches. The ablation study is well done: this part of the paper is split into categories and each one is devoted to the specific part of the pipeline.

**Weaknesses:**

Despite the strengths of the work, there are a number of questions about it. There is a method of missing parts completion in section 3.2. Many side methods like Zero 1-to-3, SDS guidance, PFNet are referenced in this section while there are very few particular discussions that are taken from these works. For example, what is the fractal approach discussion taken from PFNet? The second question is: what is the point of applying differentiable quantization to set the gaussians' opacity? It seems that there is a quite simple solution. Another weakness of the proposed work is the absence of the possibility to test the pipeline. The source code that aggregates the mentioned methods can clarify the approach that authors suggest.

**Questions:**

It might be interesting to compare the density of completed point clouds and compare it with GT density. What is the reason for the density difference? In the section 4.4. of ablation study we can see that point cloud extraction causes a kind of separate clusters. What is the reason?

It is recommended to provide a possibility to test the pipeline on a custom dataset. Moreover, it is recommended to split Figure 2 into two-three separate pictures with clarified steps description. The united scheme makes understanding the pipeline hard.

---

> ### Author Response · Authors · 2024-11-22
> **Response to Reviewer y9Sz (Part 1)**
>
> **Q1**: Details about side methods like Zero 1-to-3, SDS guidance, PFNet are referenced in this section while there are very few particular discussions that are taken from these works. For example, what is the fractal approach discussion taken from PFNet?
>
> **A1**: Sorry for the confusing parts. Zero 1-to-3 is a diffusion model trained to predict a 2D image from any specified camera pose relative to a given 2D input image, ensuring consistency with plausible 3D shapes.
> Score Distillation Sampling (SDS) guidance is typically used to align rendered 2D images from 3D representations with the distribution learned by a pre-trained diffusion model. By applying SDS guidance from Zero 1-to-3, we constrain the images rendered from our 3D Gaussians to align more closely with realistic 3D shapes.
> The Fractal approach, inspired by PFNet, focuses on completing only the missing parts of an incomplete input model while preserving existing regions, instead of directly generating fully completed point clouds like others. Following Fractal approach, we optimize $G_m$ to reconstruct the missing parts while freezing $G_{in}$, initialized from the original partial input, to retain existing shapes.
> We have added some details to the above mentioned sides methods in the Related Works, Sec.2.
>
> **Q2**: What is the point of applying differentiable quantization to set the gaussians' opacity? It seems that there is a quite simple solution.
>
> **A2**: Yes, another alternative is to constrain the opacity with loss to force it towards 0 or 1. If we set the optimized opacity attribute as $G_m^o$, such a loss can be described as
> $L_{opa} = w \cdot min(G_m^o, 1-G_m^o),$
> where $w$ denotes the corresponding weight term. $G_m^o$ is between $0 \sim 1$. Below, we present a comparisons between $L_{opa}$ with different $w$ values, and our quantization method mentioned in Sec.3.2 of the original paper. The evaluation is conducted on the synthetic objects following Sec. 4.1.
>
> |                | $w$=0.1        | $w$=1.0        | $w$=20.0 |  $w$=50.0   |$w$=100.0   |    Ours    |
> |----------------|----------|-----------|---------|------|------|--------|
> | CD             | 1.52       | 1.51       | 1.52      | 1.63| 1.66    | **1.43** |
> | EMD            | 1.80       | **1.78**   | 1.83      | 1.89| 1.90    | 1.88   |
>
> We can see that our quantization has better promotions for CD metric, while $L_{opa}$ brings more improvements on EMD. However, we can also observe that $w$ will obviously affect  the performances of $L_{opa}$, which means we may need to adjust it repeatedly to get the best setting.
> In contrast, our quantization method does not introduce hyper-parameters. Only $\delta$ in Eq.2 is always set as a constant 0.01 because smaller value will make Gaussian primitives fail to be optimized. In this sense, we think that our quantization method may be more general to use. These two methods can also be selected according to the specific application scenarios.
>
> **Q3**: What is the reason for the density difference? In the section 4.4. of ablation study we can see that point cloud extraction causes a kind of separate clusters. What is the reason?
>
> **A3**: Simply speaking, the separate clusters with different densities observed in the extracted results of Sec. 4.4 arise from the differing densities of $G_{in}$and $G_m$, as shown in Fig. 2.
>
> Initially, the optimized 3D Gaussians are distributed below, on, and above the actual surfaces. To address this, we introduce **Gaussian Surface Extraction**, which identifies and retains only the Gaussians located precisely on the surface.
>
> As described in Algorithm 1, we extract surface points by preserving only the first 3D Gaussian observed in each pixel with an opacity value greater than 0.5. When initializing $G_m$with the same number of partial points as in $P_{in}$, many Gaussians in $G_m$ that are not on the surface are removed during this process. Since $G_{in}$ is initialized directly from $P_{in}$, its points remain perfectly preserved, as they are all precisely on the surface.
>
> As a result, the extracted surface points $P_{surf}$, which are composed of points from $G_m$ and $G_{in}$, exhibit naturally sparser point densities in regions contributed by $G_m$ compared to those from $G_{in}$. While increasing the number of initialized 3D Gaussians in $G_m$ can enhance point density in specific regions, it also significantly increases computational costs during optimization. In this context, our **Grid Pulling** method offers a more computationally efficient solution for achieving uniform results.

---

> ### Author Response · Authors · 2024-11-22
> **Response to Reviewer y9Sz (Part 2)**
>
> **Q4**: About the source codes to test a custom dataset.
>
> **A4**: Thank you for your interest to our work! Due to the limited time of the rebuttal period, we do not have time to clean up our codes or make demos. But we promise we will make our codes open sourced once we have done clean up the codes.
>
> **Q5**: Moreover, it is recommended to split Figure 2 into two-three separate pictures with clarified steps description.
>
> **A5**: Thank you for your suggestion. We have carefully considered this suggestion, but splitting Figure 2 into three separate figures would greatly increase the paper length, potentially exceeding the 10-page limit. Instead, we have revised Figure 2 to improve clarity as following:
> 1.  A simplified description of the entire pipeline has been added at the bottom, along with arrows to indicate the sequence of modules.
> 2.  The caption has been adjusted to clearly correspond to the different modules in Figure 2.
>
> Please feel free to propose any further suggestions.

---

### Official Review · Reviewer_Xxcz · 2024-11-03

**Soundness:** 3
**Presentation:** 3
**Contribution:** 3
**Rating:** 6
**Confidence:** 4

**Summary:**

The paper introduces a test-time point cloud completion framework that leverages 2D diffusion priors. Unlike previous methods, which require manually provided information such as text prompts, the proposed method conditions the diffusion model on a reference image from the partial observation. After initializing the partial Gaussians and rendering the reference image, the framework optimizes the parameters of the Gaussians representing the 'missing' parts and then extracts the completed point cloud. The approach is validated on synthetic and real-world scanned point clouds.

**Strengths:**

a) The problem of completing partial point clouds in a generalizable way is important, as existing methods mostly yield good results only on in-domain samples. b) 2D/3D priors from diffusion models are well-suited for achieving generalization. c) The paper is well-written, the problem is clearly specified and the solution is well presented.

**Weaknesses:**

a) As the incompleteness of the partial point cloud increases, the diffusion model may produce different completion results at each optimization. However, the multi-modal nature of the approach is not analyzed. For a small set of shapes, multiple runs can be performed on the same partial input, and the resulting completions can be evaluated using multi-modal metrics such as TMD, UHD, MMD. b) The experiment in A.5 could be performed on observations with varying levels of incompleteness, rather than artificially eliminating points, to better demonstrate the method’s actual performance. For instance, depth maps could be used to obtain incomplete observations by merging points from a different number of views. c) The paper does not include a comparison with recent related work such as [1, 2]. The authors are suggested to compare ComPC with also these recent baselines to better evaluate its completion and generalization performance.

[1] Chu, R., Xie, E., Mo, S., Li, Z., Nießner, M., Fu, C. W., & Jia, J. (2024). Diffcomplete: Diffusion-based generative 3d shape completion. Advances in Neural Information Processing Systems, 36.
[2] Yan, X., Lin, L., Mitra, N. J., Lischinski, D., Cohen-Or, D., & Huang, H. (2022). Shapeformer: Transformer-based shape completion via sparse representation. In Proceedings of the IEEE/CVF Conference on Computer Vision and Pattern Recognition (pp. 6239-6249).

**Questions:**

All of my questions are listed in the weaknesses section, and I may adjust the rating if they are well addressed.

---

> ### Author Response · Authors · 2024-11-22
> **Response to Reviewer Xxcz (Part 1)**
>
> **Q1**: Evaluation on multi-modal metrics such as TMD, UHD, MMD.
>
> **A1**: To evaluate the multi-modal metrics of our diffusion-based method, we follow [1] to compute TMD, UHD, and MMD. We perform four repeated optimizations of both our method and SDS-complete on the in-domain categories of Redwood, as described in Sec. 4.2, to measure these metrics. The results are presented below.
>
> | Methods         | Metrics | Table         | Exe-Chair     | Out-Chair     | Old-Chair      | Aver          |
> |------------------|---------|---------------|---------------|---------------|----------------|---------------|
> | **SDS-Complete** | TMD     | 1.26          | 1.70          | 1.18          | 1.25           | 1.35          |
> |                  | UHD     | 9.31          | 10.39         | 10.63         | 16.67          | 11.75         |
> |                  | MMD     | **1.27**      | 1.66          | 1.86          | 2.11           | 1.73          |
> | **Ours**         | TMD     | **0.57**      | **0.53**      | **0.42**      | **0.65**       | **0.54**      |
> |                  | UHD     | **8.47**      | **4.73**      | **8.64**      | **12.02**      | **8.47**      |
> |                  | MMD     | 1.47          | **1.04**      | **1.28**      | **1.42**       | **1.30**      |
>
> We can see that our method still performs better on these metrics, confirming it outperforms existing diffusion-based test-time method SDS-Complete in the comprehensive performances of multiple completions.
>
> [1] Diffusion-sdf: Conditional generative modeling of signed distance functions
>
> **Q2**: Evaluation on multiple incompleteness levels;
>
> **A2**: Thank you for your suggestion. Here, we use synthetic objects from Sec. 4.1 to construct evaluation sets with varying levels of incompleteness. Specifically, we initialize the first virtual camera at a pose of $elevation = 0^\circ$, $azimuth = -140^\circ$, and $fov \approx 80^\circ$. Additional virtual cameras are placed along the azimuth at 15° intervals. By merging 1, 3, and 7 consecutive depth maps, we generate partial point clouds with different levels of incompleteness. These data are used for comparison experiments.
> The quantitative results are presented below.
> | Level | PoinTr (CD/EMD) | Seedformer (CD/EMD) | PointAttN (CD/EMD) | SVDFormer (CD/EMD) | ShapeFormer (CD/EMD) | AdaPoinTr (CD/EMD) | Ours (CD/EMD)       |
> |-------|------------------|---------------------|---------------------|---------------------|-----------------------|---------------------|---------------------|
> | 1     | 3.77/5.13        | 4.16/6.02          | 5.52/6.29          | 4.63/5.08          | 3.30/4.07            | 5.33/5.82          | **1.86/2.01**       |
> | 3     | 3.61/5.10        | 3.92/5.93          | 5.45/6.28          | 4.36/5.02          | 3.42/4.26            | 5.28/5.82          | **1.76/2.04**       |
> | 7     | 3.06/4.95        | 3.48/5.72          | 5.18/6.13          | 4.16/4.95          | 3.00/3.77            | 5.26/5.85          | **1.48/1.87**       |
>
> The mentioned quantitative results have been added to Table 10 in Sec. A.6 of the appendix, along with qualitative comparisons in Fig. 15. As shown, using more depth maps to construct the partial input enhances the completed results by providing finer details. Notably, our method consistently outperforms other approaches in this setting.

---

> ### Author Response · Authors · 2024-11-22
> **Response to Reviewer Xxcz (Part 2)**
>
> **Q3**: Comparisons with recent works [1,2].
>
> [1] Diffcomplete: Diffusion-based generative 3d shape completion.
> [2] Shapeformer: Transformer-based shape completion via sparse representation.
>
> **A3**:  Thank you for your feedback. Unfortunately, we did not find the pre-trained models for DiffComplete [1]. Due to the limited time of the rebuttal period, it is also difficult to re-train it for further comparisons. However, we have included comparisons with ShapeFormer [2] as detailed below.
> Here is the result on synthetic objects:
> | **Object**   | Horse      | MaxPlanck  | Armadillo  | Cow        | Homer      | Teapot     | Bunny      | Nefertiti  | Bimba      | Ogre       | **Average** |
> |--------------|------------|------------|------------|------------|------------|------------|------------|------------|------------|------------|------------|
> | **Metrics**  | CD/EMD     | CD/EMD     | CD/EMD     | CD/EMD     | CD/EMD     | CD/EMD     | CD/EMD     | CD/EMD     | CD/EMD     | CD/EMD     | CD/EMD     |
> | **ShapeFormer** | 4.17/5.38 | 3.48/4.49  | 3.76/4.68  | 4.53/5.29  | 2.27/2.84  | 2.55/2.86  | 4.52/4.44  | 3.09/3.87  | 5.00/5.85  | 3.39/4.69  | 3.68/4.44  |
> | **Ours**      | **0.96/1.32** | **1.23/1.53** | **2.49/4.05** | **1.45/1.64** | **1.34/1.76** | **0.99/1.22** | **1.43/1.78** | **1.81/2.20** | **1.39/1.64** | **1.22/1.67** | **1.43/1.88** |
>
> Here is the result on Redwood dataset:
> | **Object**   | Table      | Exe-Chair  | Out-Chair  | Old-Chair  | Average    | Vase       | Off Can    | Vespa      | Tricycle   | Average    |
> |--------------|------------|------------|------------|------------|------------|------------|------------|------------|------------|------------|
> | **Metrics**  | CD/EMD     | CD/EMD     | CD/EMD     | CD/EMD     | CD/EMD     | CD/EMD     | CD/EMD     | CD/EMD     | CD/EMD     | CD/EMD     |
> | **ShapeFormer** | 3.48/5.67 | 3.41/5.32  | 3.87/6.93  | 3.00/4.07  | 3.44/5.50  | 4.79/6.50  | 2.96/3.89  | 3.21/4.20  | 3.21/4.20  | 4.01/5.43  |
> | **SDS-Complete** | **1.35/2.30** | 1.96/2.65  | 2.51/3.92  | 2.77/3.77  | 2.15/3.16  | 3.00/5.25  | 3.79/4.28  | 3.36/5.73  | 3.18/3.49  | 3.33/4.69  |
> | **Ours**      | 1.67/3.11 | **1.04/1.39** | 1.28/1.73  | **1.42/1.87** | **1.35/2.03** | **2.94/4.63** | **3.51/3.86** | **1.39/2.27** | 2.42/**1.94** | **2.57/3.17** |
>
> Our method demonstrates superior performance, while ShapeFormer [2], like other fully-supervised methods, may be constrained by its training dataset.
> We have included the corresponding references of [1] and [2] in Sec. 2.2 and updated Table 1 and Table 2 in the original paper accordingly.

---

> > ### Comment · Reviewer_Xxcz · 2024-11-25
> >
> > Thank you for responding to my questions, the rebuttal addresses most of my concerns. I have increased my rating.
> >
> > A1: For TMD, higher values should indicate better performance, as it measures diversity. However, it seems like the lower values are assumed to be better in your table (bolded values). This should be corrected in the final version of the paper.

---

> > > ### Author Response · Authors · 2024-11-25
> > > **Response to Reviewer Xxcz**
> > >
> > > Thank you for your comment! We are encouraged that our response has resolved most of your concerns. We will revise the marking for TMD and include it in the final version of the paper. While our method exhibits a lower TMD, this actually indicates that it converges steadily toward the ground truths, which can be considered a positive outcome for the task of 3D point cloud completion. We will explicitly clarify this point in the revised version.

---

### Official Review · Reviewer_tTgt · 2024-11-10

**Soundness:** 3
**Presentation:** 3
**Contribution:** 3
**Rating:** 6
**Confidence:** 4

**Summary:**

This paper addresses the problem of 3D point cloud completion from partial and noisy data, specifically in an open-set setting, where point clouds from unseen categories are completed. The proposed approach introduces a novel, training-free framework that leverages 3D Gaussian Splatting (3DGS) representations and pre-trained 2D diffusion models for guiding the completion process.

First, a Partial Gaussian Initialization (PGI) module estimates a reference viewpoint, $V_p$, that maximizes coverage of the input partial point cloud, $P_{in}$. This viewpoint is used to initialize partial 3D Gaussians, $G_{in}$, and render an image $I_{in}$. The Zero-shot Fractal Completion (ZFC) module then combines $G_{in}$ with another set of 3D Gaussians, $G_m$, which is generated from a noisy variant of the input point cloud, $P_m$. Using the Zero-1-to-3 diffusion model, $G_m$ is refined via Score Distillation Guidance based on the reference viewpoint image $I_{in}$, which optimizes the Gaussians based on a randomly rendered image $I_i$.  A geometric preservation constraint is also introduced by measuring the Chamfer Distance between the input point cloud and the predicted point cloud, $P_{pre}$the predicted one $P_{pre}$ under the reference viewpoint $V_p$.

Finally, the Point Cloud Extraction (PCE) module extracts surface points, $P_{surf}$, from the 3D Gaussian centers, $G_{all} = {G_{in}, G_m}$, excluding those with low opacity and selecting the ones that are visible from a set of uniform camera poses. To address the non-uniform density of $P_{surf}$, a Grid Pulling step is applied that promotes uniform alignment of $P_{surf}$ along the underlying surface. The framework is evaluated on synthetic and real-world point clouds for the point cloud completion task, achieving superior or comparable performance w.r.t. existing methods.

**Strengths:**

The proposed framework generates complete point clouds that accurately capture the underlying surface, entirely bypassing the need for training on extensive 3D shape categories. By leveraging a 2D diffusion model to synthesize novel views from a specific input viewpoint, it also removes the need for manual text prompts. Furthermore, this approach achieves superior or comparable performance to other test-time frameworks (e.g., SDS-complete) while significantly reducing computational time. Finally, the adaptation of the NeuraPull approach, using near and far points from the 3D bounding box of the $P_{surf}$ point cloud, while also incorporating geometric details from the input partial point cloud, is able to produce uniformly-sampled point clouds.

**Weaknesses:**

Although this method claims to rely solely on 3D coordinates, as required by the point cloud completion task, it implicitly introduces additional information by incorporating the 3D normal of each point to encode the color map $G_{in}^c$. This additional input signal, which is not utilized by other methods such as PCN or SDS-complete, may introduce an advantage, as evidenced by its significant impact on performance shown in Table 3 and Figure 7. Furthermore, the method heavily depends on the input partial point cloud throughout all stages: in the ZFC module, the SDS guidance is conditioned on the reference viewpoint image derived from the input partial point cloud, and a geometric preservation constraint is based on the original point cloud. In the PCE module, the initial point cloud is merged with $P_{pull}$ in the Grid Pulling step, further reinforcing reliance on the input partial point cloud.

**Questions:**

- As noted in the **Weaknesses** section, this method incorporates the input point cloud across all stages of the pipeline. It would be beneficial to examine how the method performs under varying occlusion levels and perturbation intensities.
- Following the above, could the method leverage multiple reference viewpoints during the ZFC step? Would incorporating additional viewpoints lead to more accurate completion of point clouds, particularly in cases with high noise levels and significant occlusions?
- Could this method benefit from adopting 3D scene representations such as VolSDF [1], as used in SDS-complete, which implicitly encodes the SDF of the underlying surface?
- Clarity:
  - It’s important to clarify that the SDS-complete method leverages VolSDF [1], rather than NeuS, to represent the 3D scene.
  - Adding a horizontal separator line after the Metrics row in Tables 1, 2, and 9 would improve readability.
  - Please also add the SeedFormer method in line 418.

[1] L. Yariv et al., “Volume Rendering of Neural Implicit Surfaces”, In Proc. NeurIPS, 2021.

---

> ### Author Response · Authors · 2024-11-22
> **Response to Reviewer tTgt (Part 1)**
>
> **Q1**: The weakness about introducing extra normal maps.
>
> **A1**: As described in the Gaussian Attributes Setting in Sec. 3.1, all normal vectors are directly **estimated** using Open3D from the original partial input point clouds. Open3D computes the normal vector for each point based purely on statistical calculations involving its neighboring points. From an overall input perspective, this process does not introduce additional signals compared to other completion baselines, where only the incomplete point cloud serves as input.
>
> **Q2**: The method introduce the input point cloud across all stages of the pipeline. It would be beneficial to examine how the method performs under varying occlusion levels and perturbation intensities.
>
> **A2**: We incorporate the input partial points at different stages to ensure the completed results remain consistent with the partial input. To evaluate the performance of our method under varying occlusion levels and perturbations, we conduct the following experiments using the synthetic point clouds described in Sec. 4.1:
>
> 1.  **Partial Point Clouds with Varying Incompleteness Levels**: We construct input point clouds with different levels of incompleteness by fusing depth maps from varying numbers of virtual cameras. Specifically, we initialize the first virtual camera at a pose of $elevation = 0^\circ$, $azimuth = -140^\circ$, and $fov \approx 80^\circ$. Additional virtual cameras are placed along the azimuth at 15° intervals. By merging 1, 3, and 7 consecutive depth maps, we generate partial point clouds with different levels of incompleteness. The quantitative results are presented below.
>
> | Level | PoinTr (CD/EMD) | Seedformer (CD/EMD) | PointAttN (CD/EMD) | SVDFormer (CD/EMD) | ShapeFormer (CD/EMD) | AdaPoinTr (CD/EMD) | Ours (CD/EMD)       |
> |-------|------------------|---------------------|---------------------|---------------------|-----------------------|---------------------|---------------------|
> | 1     | 3.77/5.13        | 4.16/6.02          | 5.52/6.29          | 4.63/5.08          | 3.30/4.07            | 5.33/5.82          | **1.86/2.01**       |
> | 3     | 3.61/5.10        | 3.92/5.93          | 5.45/6.28          | 4.36/5.02          | 3.42/4.26            | 5.28/5.82          | **1.76/2.04**       |
> | 7     | 3.06/4.95        | 3.48/5.72          | 5.18/6.13          | 4.16/4.95          | 3.00/3.77            | 5.26/5.85          | **1.48/1.87**       |
>
> As shown in the table, more complete partial inputs, constructed from additional depth maps, result in improved completion outcomes.
> The qualitative comparisons are also provided in Fig. 15 of Sec. A.6 in the appendix. Notably, our method consistently outperforms other approaches under these settings.
>
> 2. **Noised Partial Point Clouds with Different Perturbation Intensities**: We generate noised partial point clouds by adding Gaussian noise with varying standard deviations (Std) to the original partial point clouds. Our method is evaluated on noised inputs with $std=\{0, 0.001, 0.005, 0.01\}$. The quantitative results are presented below.
>
> | Std    | PoinTr (CD/EMD) | Seedformer (CD/EMD) | PointAttN (CD/EMD) | SVDFormer (CD/EMD) | ShapeFormer (CD/EMD) | AdaPoinTr (CD/EMD) | Ours (CD/EMD)       |
> |--------|------------------|---------------------|---------------------|---------------------|-----------------------|---------------------|---------------------|
> | 0      | 4.10/5.63        | 4.21/5.93          | 5.72/6.54          | 4.79/5.29          | 3.68/4.44            | 5.59/6.04          | **1.43/1.88**       |
> | 0.001  | 4.15/5.59        | 4.17/5.91          | 5.76/6.55          | 4.73/5.22          | 3.65/4.52            | 5.59/6.04          | **1.53/1.87**       |
> | 0.005  | 4.24/5.83        | 4.31/6.11          | 5.75/6.59          | 4.92/5.52          | 4.03/4.89            | 5.65/6.19          | **2.02/2.25**       |
> | 0.01   | 4.16/5.85        | 4.34/6.22          | 5.62/6.73          | 4.86/5.75          | 4.06/5.00            | 5.57/6.44          | **3.18/4.07**       |
>
> While the performance of our method decreases as noise increases, it consistently outperforms existing approaches. Qualitative results in Fig. 14 of Sec. A.5 in the appendix demonstrate that even with a noise standard deviation of 0.01—causing noticeable blurring in the input partial points—our method can still recover the overall contour effectively. This confirms that our method exhibits a degree of robustness to noise.
>
> Moreover, our primary contribution in this work is the development of a practical framework for leveraging 2D diffusion priors in 3D point cloud completion. Comparisons on real scans from the Redwood dataset further validate the effectiveness of our method in handling real-world data. The robustness against noise is not the core of this work, we can further explore it in our future research.

---

> > ### Comment · Reviewer_tTgt · 2024-11-26
> >
> > I thank the authors for their response.
> >
> > Regarding whether point normals are an additional input signal, I was already aware from my pre-rebuttal review that these were estimated using Open3D by computing the principal axis of a local neighborhood through PCA. However, even with these estimated normals, they can be quite robust when the input point cloud is noise-free (no perturbation). This is why I consider them an additional input. This robustness is also evident in the ablation under various perturbation levels. While the method outperforms competing approaches, the CD/EMD metrics double under moderate perturbation ($\sigma = 0.01$), which contrasts with most other methods whose performance remains more consistent.
> >
> > For the ablation under varying completeness levels, I may not have been clear in my original review, but I would like to see the method's performance on more incomplete partial point clouds. From Figure 15, it seems the method is tested on point clouds that are as complete or even more complete compared to those in other qualitative evaluations, such as Figures 5 and 6.

---

> ### Author Response · Authors · 2024-11-22
> **Response to Reviewer tTgt (Part 2)**
>
> **Q3**: Could the method leverage multiple reference viewpoints during the ZFC step?
>
> **A3**: It may be a little difficult. For the task of point cloud completion, all we can get is a partial point cloud. To complete it, we select a **reference viewpoint**—the viewpoint from which the partial point cloud appears the most complete.
> As illustrated in $I_{in}$ of Fig. 2, this viewpoint makes the partial points **look like a complete point cloud at most**, allowing us to extract comprehensive geometric priors from the diffusion model using SDS guidance. For real scanned point clouds, the reference viewpoint  should actually be close to the sensor or camera's perspective.
> In this framework, the reference viewpoint is unique to our method. Adding additional viewpoints may degrade performance, as they could introduce more incomplete regions from the partial input into the reference image, ultimately disrupting the SDS guidance.
>
> **Q4**: Could this method benefit from adopting 3D scene representations such as VolSDF [1], as used in SDS-complete, which implicitly encodes the SDF of the underlying surface?
>
> **A4**: In this work, we adopt 3D Gaussian Splatting as the 3D representation for introducing 2D diffusion priors due to its strong coupling with point clouds. By defining color-related attributes, point clouds can be transformed into 3D Gaussians for differentiable rendering, while the optimization of 3D Gaussians directly manipulates the shape of the point clouds. Our experiments demonstrate that Gaussian Splatting-based frameworks are effective for point cloud completion.
>
> To explore the usage of continuous 3D representations, such as VolSDF, one potential approach is to co-optimize VolSDF through the centers of 3D Gaussians and the 2D diffusion priors during Zero-shot Fractal Completion (ZFC). The final point cloud could then be sampled directly from VolSDF, eliminating the need for Point Cloud Extraction (PCE). This could potentially improve completion performance by avoiding precision loss associated with PCE for 3D Gaussians. However, incorporating another 3D representation would likely result in significantly increased computational costs.
> Due to the limited time of the rebuttal period, we cannot pursue this attempt at this moment, but it remains a promising avenue for future work.
>
> **Q5**: Clarity about the citations and other contents.
>
> **A5**: Thank you for your valuable feedback. We sincerely apologize for the incorrect reference in our original submission. We have thoroughly revised the paper to address all the issues mentioned.

---

> ### Author Response · Authors · 2024-11-27
> **Response to Reviewer tTgt (Part 1)**
>
> Thank you for your insightful comments! Here, we will present response to your concerns point-by-point:
>
> **Q1**: Clarity about the using of normal map;
>
> **A1**: On the one hand, as stated in our previous response, all the information required for normal estimation is derived directly from the original point clouds. Although existing baselines based on supervised learning do not explicitly incorporate the estimated normal maps as input, their networks have learned to implicitly extract features from the input point clouds during training. The input of our framework is just the same partial point cloud as them, where the normals can actually be regarded as our manually extracted features. From this perspective, we think that normals should not be considered as additional input.
>
> On the other hand, while the normal map is primarily used for colorizing the point clouds, its role is **important but not irreplaceable**.
>
> In Table 3 of the original paper, we conducted an ablation study on the Redwood dataset, which shows that colorization using **xyz coordinates** achieves slightly lower but comparable performance to using the normal map.
> In this section, we provide a comparison on synthetic data, as shown in the table below, between these two colorization strategies.
>
> || PoinTr (CD/EMD) | Seedformer (CD/EMD) | PointAttN (CD/EMD) | SVDFormer (CD/EMD) | ShapeFormer (CD/EMD) | AdaPoinTr (CD/EMD) | Ours (Coor) (CD/EMD) | Ours (Normal) (CD/EMD) |
> |-----|------------------|---------------------|---------------------|---------------------|-----------------------|---------------------|-----------------------|------------------------|
> | Aver      | 4.10/5.63        | 4.21/5.93          | 5.72/6.54          | 4.79/5.29          | 3.68/4.44            | 5.59/6.04             | 1̲.̲7̲9̲/̲2̲.̲0̲1      | **1.43/1.88**         |
>
> The results indicate that our framework with coordinate-based colorization still outperforms other baseline methods, though slightly inferior to normal map-based colorization.
>
> This confirms that the use of normal maps is **not the sole reason** for our method's superior performance over other methods.  Instead, normal maps just act as a colorization technique that enhances completion performance by better describing surface shapes in the rendered reference image. More effective colorization techniques could be explored in future work to further enhance the completion performance.
>
> **Q2**: Clarity about the robustness.
>
> **A2**: We acknowledge that robustness to noise may be a potential limitation of our approach. We will discuss about it more thoroughly in the limitation section of our paper. However, this limitation does not diminish the value of our method.
>
> Firstly, while other baselines may appear to maintain consistent performance under different noise, their results—illustrated in Fig. 14 of the revised paper—are **clearly inferior** from the outset. In contrast, our method produces significantly more applicable completion results.
>
> Secondly, our experiments on real scans from RedWood dataset in Sec.4.2 shows that our method can perform superior in practical applications. It has confirmed that our method has **robustness to real world data**.
>
> Thirdly, as mentioned in our previous response, robustness to noise is **not the primary focus** of this work. Instead, our core contribution lies in designing a practical framework for 3D point cloud completion that does not rely on training with specific datasets, additional prompts, but solely on 2D diffusion priors.
>
> The robustness of this framework could potentially be enhanced  by integrating additional denoising modules, such as advanced outlier removal methods for noisy point clouds. However, such extensions fall outside the scope of this work. We can consider about them in our future research.

---

> ### Author Response · Authors · 2024-11-27
> **Response to Reviewer tTgt (Part 2)**
>
> **Q3**: Comparison on more incomplete data;
>
> **A3**:  In this work, we follow the setup introduced by PCN [1], where incomplete point clouds are derived from **single-frame depth maps**. Most existing point cloud completion methods, such as [2, 3], do not consider about more incomplete cases because a single-frame depth map is typically the smallest unit for practical sensor-based collection. Therefore, as shown in our previous responses and Fig. 15, we design varying levels of incompleteness by fusing different numbers of depth maps, reflecting a more practical scenario.
>
> To further evaluate the robustness of our method, we also present comparisons on highly incomplete point clouds by removing points from the partial inputs, as detailed in the table below. Specifically, 30%, 50%, and 70% of points are randomly erased from the original partial point clouds.
>
> | Removed | PoinTr (CD/EMD) | Seedformer (CD/EMD) | PointAttN (CD/EMD) | SVDFormer (CD/EMD) | ShapeFormer (CD/EMD) | AdaPoinTr (CD/EMD) | Ours (CD/EMD)       |
> |---------|------------------|---------------------|---------------------|---------------------|-----------------------|---------------------|---------------------|
> | 0      | 4.10/5.63        | 4.21/5.93          | 5.72/6.54          | 4.79/5.29          | 3.68/4.44            | 5.59/6.04          | **1.43/1.88**       |
> | -30%    | 4.07/5.51        | 4.20/5.97          | 5.76/6.58          | 4.77/5.23          | 3.85/4.66            | 5.56/6.02          | **1.55/1.87**       |
> | -50%    | 4.15/5.57        | 4.21/5.98          | 5.75/6.55          | 4.78/5.04          | 3.86/4.46            | 5.65/6.15          | **1.68/2.02**       |
> | -70%    | 4.13/5.64        | 4.19/5.99          | 5.81/6.53          | 4.85/5.22          | 3.89/4.81            | 5.72/6.07          | **1.78/2.18**       |
>
> The results demonstrate that the performance of our method remains relatively stable across different levels of incompleteness. A possible reason is that removing points does not significantly alter the overall surface shapes, and thus does not greatly affect the colorized reference image, unlike direct noise perturbations.
>
> **References:**
> 1. PCN: Point Completion Network
> 2. TopNet: Structural Point Cloud Decoder
> 3. DiffComplete: Diffusion-Based Generative 3D Shape Completion
>
> Please feel free to present any further comments or questions.

---

> ### Author Response · Authors · 2024-12-02
> **Follow-up before Discussion Period Deadline**
>
> Dear Reviewer  tTgt,
>
> Thank you again for your insightful comments! Since the deadline of the reviewer-author discussion period is approaching **today**, we wonder if you have any further questions about our work, so that we can response to them timely.
>
> In the previous comment, we presented responses to the proposed **problems**:
>
> (1) **About the normal map**: Does our method outperform other baselines due to the advantage provided by the additional input normal map?
>
> (2) **About the robustness**: although our method maintains better performance under varying noise levels, its performance decreases faster than other methods.
>
> (3) **About more incomplete data**: how would the performance change with more incomplete input?
>
> We would quite appreciate it if you could provide any further feedback to help us further improve our work.
>
> Sincerely,
>
> Authors

---

### Author Response · Authors · 2024-11-22
**Global response to all reviewers**

We sincerely thank all reviewers for their valuable feedback. We are encouraged that all reviewers acknowledge the strong motivation and solid performance of our method. We apologize for any confusion caused by missing comparisons, citations, unclear definitions, or other issues. In this rebuttal, we have carefully replied to each concern point by point. All corresponding revisions to the raised questions have been highlighted in red in the revised paper. We hope our responses effectively address your concerns. Please feel free to share any additional questions or suggestions regarding this work.

---

### Meta-Review · Area_Chair_izpK · 2024-12-21

**Metareview:**

This paper introduces an approach for point cloud completion. Instead of data-driven approaches based on manually organized training sets, the proposed approach utilizes 2D diffusion priors. The proposed approach is not affected by the conventional 3D dataset, which is often manually organized or biased toward certain categories. The 3D point clouds are represented with 3D Gaussians and completed with 2D diffusion priors, and the final point clouds are extracted uniformly. Given the compelling results and detailed description of the proposed approach, AC confirms that this paper can shed light on the possibility of the practical use of diffusion priors for dense prediction tasks. However, handling non-object-centric point clouds or recovering detailed surfaces seems to be a future work of this approach.

**Additional Comments On Reviewer Discussion:**

In summary, the reviewers gave positive scores for the paper acceptance after the discussion phase. All reviewers except for the reviewer, p7jk, provided constructive comments. The authors provided solid feedback on the reviewers' questions regarding missing comparisons, citations, and unclear discussions.

More specifically, the reviewer tTgt initially gave negative scores, with concerns regarding the use of nearly perfect normal maps, robustness under the various occlusion levels and perturbations, viewpoint selections, and the possibility of using signed distance functions. The authors faithfully provided extra experimental results to supplement their explanation. The reviewer tTgt raised the score. The reviewer Xxcz requested additional evaluation with multi-modal metrics, multiple incompleteness levels, and comparison with more recent approaches. The reviewer provided solid feedback. The reviewer y9Sz provides a review with a strong score. The reviewer requested additional discussion regarding SDS guidance and how the proposed approach differs from Zero 1-to-3. The reviewer also asked about the opacity of Gaussians and the visual inconsistency of the results appearing in Sec 4.4. The authors answered these questions, including other minor requests. The reviewer, p7jk, briefly reviews how the proposed approach can generate a surface. Since the reviewer p7jk did not give the ordinal review and the question does not address the key point of the paper, AC downweights the comments by the reviewer.

---

### Decision · Program_Chairs · 2025-01-22

Accept (Poster)